# REVISUAL-R1: ADVANCING MULTIMODAL REASONING FROM OPTIMIZED COLD START TO STAGED REINFORCEMENT LEARNING

**Shuang Chen**[1]***Hangyu Guo**[2]***Zhaochen Su**[3] , **Yafu Li**[4], **Yulun Wu**[1], **Jiacheng Chen**[4],
**Jiayu Chen**[2], **Weijie Wang**[1], **Zhiyuan Feng**[1], **Xiaoye Qu**[4]†, **Yu Cheng**[5]†

[1] Zhejiang University [2] Fudan University [3] Soochow University
[4] Shanghai AI Laboratory [5] The Chinese University of Hong Kong

**Code:** https://github.com/CSfufu/Revisual-R1

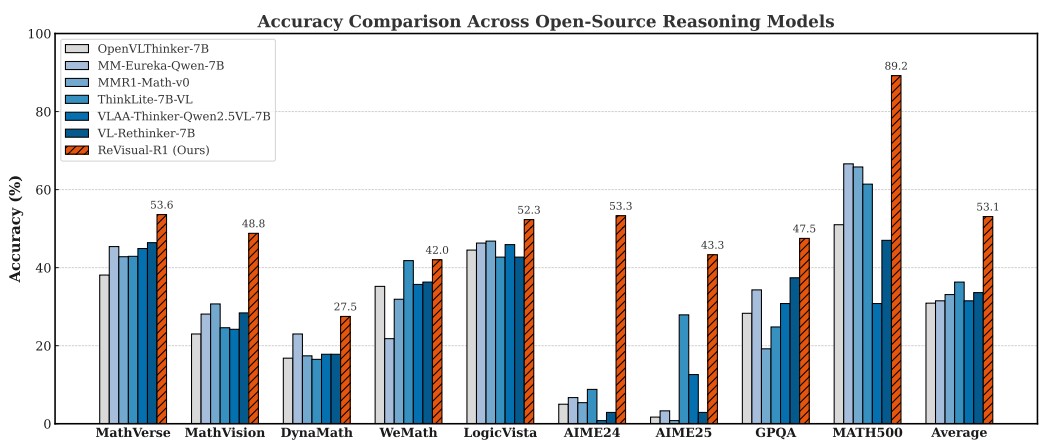

**Figure 1:** Overall performance across five multimodal reasoning benchmarks (MathVerse, MathVision, DynaMath, WeMath, and LogicVista) and four textual reasoning benchmarks (AIME24, AIME25, GPQA, and MATH500). Our **ReVisual-R1** achieves better performance than existing works.

## ABSTRACT

Inspired by the remarkable reasoning capabilities of Deepseek-R1 in complex textual tasks, many works attempt to incentivize similar capabilities in Multimodal Large Language Models (MLLMs). However, they still struggle to activate complex reasoning. In this paper, rather than examining multimodal RL in isolation, we delve into current training pipelines and identify three crucial phenomena: 1) Effective cold start initialization is critical for enhancing MLLM reasoning. Intriguingly, we observe that initializing with carefully selected text data alone can lead to performance surpassing many recent multimodal reasoning models, even before multimodal RL. 2) Standard GRPO applied to multimodal RL suffers from gradient stagnation, which degrades both training stability and performance. 3) A final text-only RL tuning stage, conducted after the multimodal RL phase, is effective at further sharpening multimodal reasoning capabilities. Capitalizing on these insights, we introduce **ReVisual-R1**, achieving a new state-of-the-art among open-source 3B and 7B MLLMs on challenging benchmarks, including MathVerse, MathVision, WeMath, LogicVista, DynaMath, and challenging AIME2024 and AIME2025. This paradigm demonstrates robust scalability, proving effective at both 7B and 3B scales. Our findings chart a new course for training powerful multimodal reasoners, demonstrating that a carefully orchestrated, multi-stage strategy is key to unlocking their full potential.

---

*Equal contributions.

†Corresponding authors.

# 1  INTRODUCTION

Recently, the field of large language models (LLMs) has witnessed significant advancements in complex cognitive reasoning (Zeng et al.; Yan et al., 2025; Zhang et al., 2025a), notably exemplified by reasoning models like DeepSeek-R1 (Guo et al., 2025a). These models successfully leveraged Reinforcement Learning (RL) to facilitate the self-emergence of intricate reasoning abilities in text-only models. A natural and ambitious next step is to extend this RL paradigm to Multimodal Large Language Models (MLLMs), with the goal of unlocking a similar leap in multimodal cognitive abilities (Huang et al., 2025b; Meng et al., 2025b; Xia et al.; Peng et al., 2025).

However, this ambition has been met with a formidable challenge. The direct application of text-centric RL techniques to MLLMs has yielded diminishing returns, suggesting that the path to multimodal reasoning is not a simple matter of architectural extension. A fundamental question arises: *how does one cultivate the abstract linguistic skill while simultaneously grounding it in the continuous, high-dimensional space of visual perception?* Naive co-training often results in a compromise, where neither modality's potential is fully realized. Motivated by this critical impasse, we undertake a rigorous dissection of the MLLM training pipeline. Our investigation uncovers three foundational insights that, when addressed in concert, chart a new and more effective course for developing powerful multimodal reasoners.

First, we observe that sufficient cold start initialization is indispensable for effectively cultivating the reasoning ability of MLLMs. Conventional cold-start phases for MLLMs often rely on simplistic visual and textual pre-training corpora (Yang et al., 2025; Huang et al., 2025b; Wang et al., 2025; Deng et al., 2025; Chen et al., 2025b; Feng et al., 2025). This initial deficit critically hinders the subsequent RL stages from eliciting sophisticated, self-critical reasoning patterns. To unlock deeper deliberative reasoning in MLLMs, an enriched cold-start initialization is therefore not merely beneficial but indispensable. Specifically, initializing with carefully selected text data that instills foundational reflective capabilities and the capacity for extended Chain-of-Thought (CoT) reasoning proves to be a powerful strategy. Intriguingly, such targeted textual initialization allows our model to surpass the multimodal reasoning performance of many recent multimodal reasoning models.

Second, we identify that the standard Group Relative Policy Optimization (GRPO) algorithm (Shao et al., 2024), commonly applied for multimodal RL, suffers from a gradient stagnation problem. This issue significantly degrades both the training stability and the ultimate performance of the multimodal RL phase. To address this fundamental limitation and improve the efficacy of multimodal RL, we propose Prioritized Advantage Distillation (PAD). PAD is designed to mitigate gradient stagnation by strategically filtering out zero-advantage samples and re-weighting informative trajectories, thereby focusing the learning process on more impactful data and improving training stability.

Third, we discover that conducting further post-training using text RL after the multimodal RL training phase can further enhance multimodal reasoning ability. This stage acts as a polishing step, sharpening the model's linguistic expression and logical consistency without eroding the newly acquired visual grounding. It consolidates the model's capabilities, leading to a synergistic whole greater than the sum of its parts.

Synthesizing these insights, we propose a principled, three-stage training curriculum that strategically sequences these learning phases: (1) a text-centric cold-start to forge a powerful reasoning engine, (2) a multimodal RL phase with PAD to ground this engine in vision, and (3) a text-only RL refinement to consolidate and polish the integrated skills. The culmination of this methodology is **ReVisual-R1**, the first 7B-parameter open-source MLLM architected around this curriculum. Extensive experiments on a suite of challenging benchmarks, including MathVerse (Zhang et al., 2024b), MathVision (Wang et al., 2024a), MathVista (Lu et al., 2023a), DynaMath (Zou et al., 2025), We-Math (Qiao et al., 2024a), and LogicVista (Xiao et al., 2024), as well as the AIME24/25 (Li et al., 2024), GPQA (Rein et al., 2024), MATH-500 (Hendrycks et al., 2021) benchmark, confirm that ReVisual-R1 significantly outperforms much larger public models. We further validate the scalability of our framework by demonstrating its effectiveness at the 3B model scale.

To summarize, our contributions are as follows:

- We challenge the conventional MLLM training paradigm by demonstrating that a text-centric, high-difficulty cold-start is the crucial, and previously overlooked, foundation for unlocking elite multimodal reasoning.

- We identify the fundamental problem of gradient stagnation in multimodal RL and propose Prioritized Advantage Distillation (PAD), a novel and effective solution that stabilizes training and enhances sample efficiency.
- We present ReVisual-R1, an open-source 7B MLLM developed through a principled three-stage curriculum. This approach uniquely cultivates deep, self-reflective reasoning and robust visual grounding, enabling ReVisual-R1 to achieve state-of-the-art performance on complex multimodal reasoning tasks, rivaling even larger or proprietary models.

## 2 PRELIMINARIES

In this section, we first formulate the task setting and key concepts in the multimodal reasoning problem. Then, we describe the base training algorithm framework used in our method.

### 2.1 MULTIMODAL REASONING FORMULATION

In multimodal reasoning tasks, the input can be represented as $x = (v, q)$, where $v$ denotes the visual content, and $q$ denotes the textual query. Our work aims to guide a MLLM to generate a multi-step, self-reflective reasoning process $t$, which ultimately assists the model in producing a solution $y$ that correctly answers the query based on the multimodal input.

Formally, we aim to learn a policy $\pi_\theta(y|x)$, parameterized by $\theta$, which maps the input question space $\mathcal{X}$ to the solution space $\mathcal{Y}$. Our objective is to optimize the model parameters such that the expected reward $r(y, x)$ over the output distribution is maximized:

$$\theta^* = \arg\max_\theta \mathbb{E}_{x \sim \mathcal{D}} \mathbb{E}_{y \sim \pi_\theta(y|x)}[r(y, x)] \tag{1}$$

where $\mathcal{D}$ represents the distribution of multimodal reasoning tasks. Similar to Deepseek R1 (Guo et al., 2025a), we mainly use rule-based reward, $r(x, y) = 1$ if $y$ is correct, otherwise $r(x, y) = 0$.

### 2.2 GROUP RELATIVE POLICY OPTIMIZATION

Group Relative Policy Optimization (GRPO) extends traditional policy optimization methods by organizing training samples into groups and optimizing policies relative to reference models within each group, offering several advantages for training language models on complex reasoning tasks.

Formally, given a batch of samples $\mathcal{B}$, GRPO divides them into $K$ groups $\{\mathcal{G}_1, \mathcal{G}_2, \ldots, \mathcal{G}_K\}$ based on certain criteria. For each group $\mathcal{G}_i$, we maintain both a policy model $\pi_\theta$ and a reference model $\pi_{\theta_{\text{ref}}}$. The GRPO objective for each group is formulated as:

$$\mathbb{E}_{x \sim \mathcal{G}_i} \mathbb{E}_{y \sim \pi_\theta(y|x)} \left[ \min \left( \frac{\pi_\theta(y|x)}{\pi_{\theta_{\text{ref}}}(y|x)} \hat{A}(x, y), \text{clip} \left( \frac{\pi_\theta(y|x)}{\pi_{\theta_{\text{ref}}}(y|x)}, 1 - \epsilon, 1 + \epsilon \right) \hat{A}(x, y) \right) \right] \tag{2}$$

where $\epsilon$ is a hyperparameter controlling the size of the trust region, and $\hat{A}(x, y)$ is the group-specific advantage function. For each input $x$ with $G$ generated responses $\{y_1, \ldots, y_G\}$ within a group, the advantage for response $y_i$ is defined as:

$$\hat{A}(x, y_i) = \frac{r(x, y_i) - \text{mean}(\{r(x, y_1), \ldots, r(x, y_G)\})}{\text{std}(\{r(x, y_1), \ldots, r(x, y_G)\}) + \epsilon} \tag{3}$$

where $\epsilon$ is a small constant for numerical stability. This relative advantage is then used within a clipped surrogate objective function. $r(x, y_i)$ represents the reward for response $y_i$ to input $x$. This advantage function measures how much better a specific response is compared to the average performance within its group, normalized by the group's reward variance.

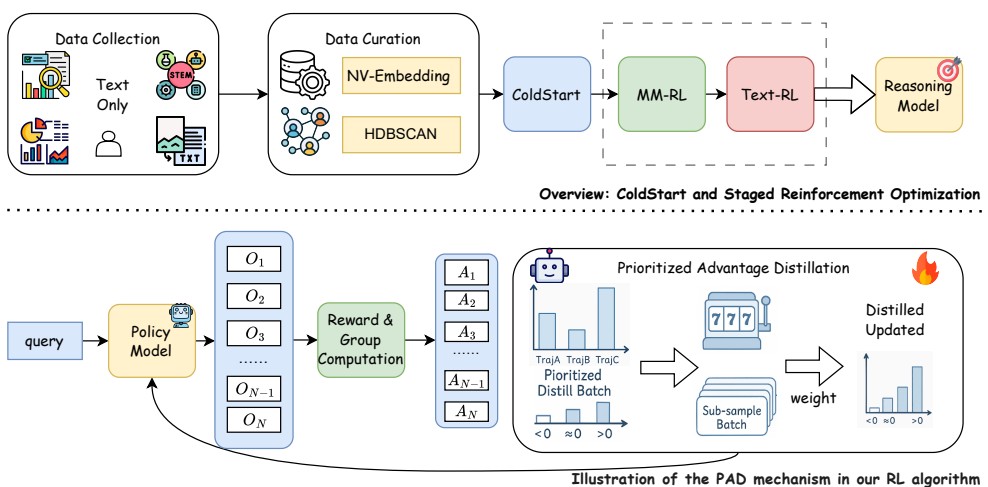

**Figure 2:** (Top): the overview of our proposed ReVisual-R1 framework. After collecting and curating data, ReVisual-R1 contains cold start and staged reinforcement learning. (Bottom): the process of our proposed prioritized advantage distillation (PAD) for multimodal reinforcement learning.

## 3 OPTIMIZED COLD START FOR MULTIMODAL REASONING

In this section, we first show an intriguing finding in the cold start of multimodal reasoning in Section 3.1, paving the way for the strategy of our data curation pipeline in Section 3.2. As shown in Figure 2, our Revisual-R1 includes a cold start stage followed by staged reinforcement learning.

### 3.1 PRELIMINARY STUDY OF COLD START

To investigate the effectiveness of cold-start training (Guo et al., 2025a), we first collect two open-source cold-start multimodal datasets, Vision-R1 (Huang et al., 2025b) and R1-One-Vision (Yang et al., 2025), along with two cold-start textual datasets, DeepMath (He et al., 2025) and OpenR1-Math (Face, 2025). Then we randomly sample 40,000 instances from these datasets to fine-tune Qwen2.5-VL-7B-Instruct (Bai et al., 2025). The fine-tuned models are subsequently evaluated on multimodal reasoning benchmarks (MathVerse and MathVision) as well as textual reasoning benchmarks (AIME24 and Math500). The experimental outcomes and average performance enhancements from the multimodal and textual cold-start datasets are illustrated in Figure 4.

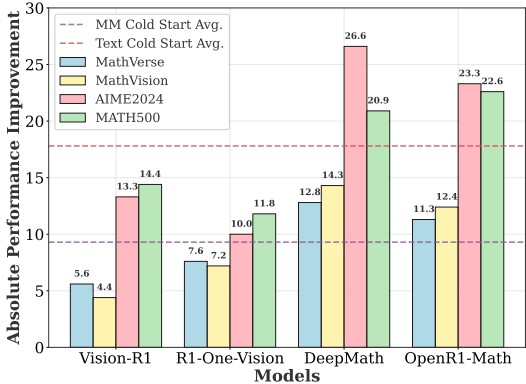

**Figure 3:** Absolute performance improvement on Qwen2.5-VL-7B-Instruct across textual and multimodal reasoning tasks. The purple and red dashed lines represent the average absolute gains of VisionR1/R1-One-Vision and DeepMath/OpenR1-Math over the baseline, respectively, across four reasoning tasks.

The results in Figure 4 reveal that models trained with text-only cold start data exhibit substantial improvements in both textual and multimodal reasoning tasks. In contrast, models trained solely on multimodal datasets, such as Vision-R1 and R1-One-Vision, show limited gains in both multimodal and textual reasoning. This suggests that the complexity and patterns presented by textual cold start data may better stimulate the models' reasoning capabilities.

To further investigate this observation, we perform an analysis using a subset of 100 examples sampled from the Vision-R1 (Huang et al., 2025b) and DeepMath (He et al., 2025) datasets. Specifically, we analyze the response lengths and pass rates of the doubao-1.5-thinking-pro-vision model (Seed

**Table 1:** Textual and multimodal reasoning datasets source of our GRAMMAR.

| | Multimodal | | | | Text-only | | | |
|---|---|---|---|---|---|---|---|---|
| Source | Samples | Source | Samples | Source | Samples | Source | Samples |
| FigureQA (Kahou et al., 2018) | 100K | Super-CLEVR (Li et al., 2023) | 30K | Big-Math-RL (Albalak et al., 2025) | 251K | GAIR_LIMO (Ye et al., 2025) | 0.8K |
| MAVIS (Zhang et al., 2025b) | 218K | TabMWP (Lu et al., 2023b) | 38K | Big-Math-RL-U | 35K | s1K-1.1 (Muennighoff et al., 2025) | 1K |
| GeoQA (Chen et al., 2021) | 5K | UniGeo (Chen et al., 2022) | 16K | OpenThoughts (Team, 2025) | 114K | OpenMathR (Moshkov et al., 2025) | 3,200K |
| Geometry3K (Lu et al., 2021a) | 2.1K | MultiMath (Peng et al., 2024) | 300K | DeepMath (He et al., 2025) | 103K | OrcaMath (Mitra et al., 2024a) | 200K |
| IconQA (Lu et al., 2021b) | 107K | | | OpenR1-220k (Face, 2025) | 220K | NuminaMath-CoT (LI et al., 2024) | 859K |

et al., 2025) on these samples. Responses to textual prompts from DeepMath averaged 8,207.76 tokens, which is substantially longer than the 821.48 tokens generated in response to multimodal prompts from Vision-R1. Moreover, the pass rate for Vision-R1 is 96.00%, whereas DeepMath achieves a pass rate of only 75.0%. These findings further indicate that current multimodal cold start datasets may lack sufficient complexity to inspire advanced reasoning capabilities of reasoning models. Therefore, in this paper, we adopt textual-only data for the cold start stage.

## 3.2 GRAMMAR: Generalized Multimodal Reasoning Dataset

Informed by Section 3.1, in this paper, we develop the Generalized Multimodal Reasoning Dataset (GRAMMAR) to enhance the generalization of reasoning capabilities in multimodal models. Specifically, the GRAMMAR dataset consists of two components. For the cold-start stage, it includes 283K diverse and complex textual samples that feature explicit reasoning paths. For reinforcement learning with a verifiable reward, it contains an additional 31K complex textual examples and 21K multimodal questions, all of which are annotated with ground truths.

As shown in Figure 2, the construction of GRAMMAR involves a multi-stage curation pipeline. We begin by amassing open-source reasoning datasets in Table 1, spanning various difficulty levels. This initial collection underwent rule-based filtering to ensure answer verifiability, excluding items like proof problems and those with difficult-to-verify ground truths. Subsequently, Qwen2.5-VL-7B-Instruct is employed for initial pruning of overly simple or complex questions. Then, Qwen2.5-VL-32B-Instruct is used to assess the remaining samples to classify them into ten difficulty levels. To maximize data diversity and minimize redundancy, we encoded questions using NV-Embedding-V2 (Lee et al., 2024), applied HDBSCAN (Campello et al., 2013) for clustering, assigned topics to clusters via Qwen2.5-7B-Instruct, and performed balanced sampling across both topics and difficulty strata.

## 4 Staged Reinforcement Optimization (SRO)

In Section 3, our data investigations and the curation of the GRAMMAR dataset highlight the necessity of high-quality, reasoning-focused data for developing advanced MLLM capabilities. In this section, we introduce Staged Reinforcement Optimization (SRO) to systematically cultivate robust reasoning and diverse competencies in MLLMs. Specifically, SRO contains two stages including multimodal RL and subsequently textual RL.

### 4.1 Stage 1: Multimodal RL

After the cold start training, the SRO framework commences with a dedicated Multimodal Reinforcement Learning (MRL) phase. This initial stage is pivotal for enabling the MLLM to ground textual concepts in visual information and execute cross-modal reasoning, primarily using the multimodal samples from our GRAMMAR dataset. We employ GRPO as the core RL algorithm for this phase. To ensure stable and effective learning, particularly when dealing with complex tasks and potentially sparse rewards common in multimodal settings, we propose a novel Prioritized Advantage Distillation (PAD) to improve gradient quality by addressing specific GRPO limitations.

### 4.1.1 Prioritized Advantage Distillation (PAD)

During multimodal RL training, we discover a significant challenge when applying GRPO in complex multimodal settings is "Gradient Stagnation". This phenomenon refers to a reduction in learning efficacy due to a predominance of near-zero advantage estimates, which is particularly acute

when dealing with sparse binary rewards. Essentially, if entire groups of generated responses yield uniform rewards (e.g., all correct or all incorrect), the resulting advantage signals become null, leading to zero policy gradients and thereby halting learning for those samples. This issue, also noted in concurrent works (Wang et al., 2025; Yu et al., 2025), can severely impede training progress. To specifically counteract gradient stagnation and enhance the efficiency of GRPO, we introduce Prioritized Advantage Distillation (PAD). PAD refines the training process by strategically focusing updates on the most informative samples within each batch, namely those exhibiting significant, non-zero advantage signals. The PAD mechanism, detailed below, operates on each batch after initial advantage estimation. It mainly contains the following three parts:

- **Per-Sequence Advantage Calculation:** Compute the absolute advantage $|\hat{A}_i|$ for each sequence $i$ in the original batch $\mathcal{B}$, representing its learning signal magnitude.
- **Effective Sample Filtering:** Form an "effective set" $\mathcal{E}$ by selecting sequences $i$ whose absolute advantage $|\hat{A}_i|$ falls within a specified informative range $[T_{low}, T_{high}]$. Critically, $T_{low} > 0$ filters out stagnant (near-zero advantage) samples, ensuring that candidates for sub-sampling provide potentially useful learning signals.
- **Prioritized Sub-sampling from Effective Set:** From this effective set $\mathcal{E}$, $k' = \min(\rho|\mathcal{B}|, |\mathcal{E}|)$ sequences are drawn to form a distilled mini-batch. Selection is prioritized based on sequences' absolute advantages ($\hat{A}_i$ for $i \in \mathcal{E}$), with the probability for selecting sequence $i$ determined by a temperature-controlled Softmax distribution:

$$\Pr(i \text{ is selected} \mid i \in \mathcal{E}) = \frac{\exp(\hat{A}_i/\tau)}{\sum_{j \in \mathcal{E}} \exp(\hat{A}_j/\tau)} \quad (4)$$

The temperature $\tau$ governs sampling concentration and is typically decayed during training (e.g., linearly from $1.0$ to $0.3$) to shift from exploration towards exploitation. This enriches the mini-batch with the most informative samples from $\mathcal{E}$.

PAD thus directly counteracts gradient stagnation via a dual mechanism: first, by filtering out stagnant samples, and second, by prioritizing updates using informative, non-zero advantages from the remaining set. This selective optimization of the learning process ensures efficient computational resource allocation towards high-value samples. Consequently, PAD leads to enhanced training stability, improved learning efficiency, and more effective acquisition of complex reasoning skills.

## 4.2 STAGE 2: TEXTUAL RL

While MRL is indispensable for grounding reasoning across visual and textual inputs, intensive MRL training can inadvertently lead to a decline in purely textual capabilities. To further elevate the model's capacity for sophisticated abstract reasoning, we integrate a subsequent Textual Reinforcement Learning (TRL) phase. This stage aims to achieve both robust linguistic fluency and advanced reasoning. Linguistic fluency is restored and enhanced by fine-tuning on high-quality, text-only corpora focused on instruction-following and conversational abilities. Simultaneously, to foster advanced reasoning, the TRL phase exposes the model to complex, text-centric problem-solving tasks. This compels the model to refine and generalize intricate reasoning patterns, articulate multi-step thought processes with greater clarity, and master linguistic nuances essential for higher-order cognition. For policy optimization during this TRL phase, we employ GRPO, augmented with our proposed PAD mechanism for efficient sample utilization.

## 5 EXPERIMENTS

### 5.1 EXPERIMENTS SETUP

**Benchmarks** To comprehensively evaluate our model's performance, we selected a diverse suite of benchmarks targeting a wide range of reasoning abilities. The evaluation suite is organized by domain: (1) Multimodal mathematical reasoning, assessed using MathVerse (Zhang et al., 2024b), MathVision (Wang et al., 2024a), MathVista (Lu et al., 2023a), DynaMath (Zou et al., 2025), and WeMath (Qiao et al., 2024b); (2) General multimodal reasoning, evaluated with LogicVista (Xiao et al., 2024), MMMU (Yue et al., 2024a), MMMU-Pro (Yue et al., 2024b), CMMMU (Zhang et al., 2024a), and MMStar Chen et al. (2024); and (3) Text-based reasoning, measured on challenging

mathematical benchmarks like AIME24/25 (Li et al., 2024) and MATH-500 (Hendrycks et al., 2021), as well as general reasoning benchmarks such as GPQA (Rein et al., 2024) and MMLU Pro (Wang et al., 2024b). Performance is reported as pass@1 accuracy for all benchmarks, with the exception of AIME24/25, for which we use average@32 accuracy.

**Baselines**  As shown in Table 2, baselines include: (1) leading closed-source models doubao-1.5-vision-pro-32k (Guo et al., 2025b), OpenAI-GPT-4o (Hurst et al., 2024), Claude-3.7-Sonnet (Anthropic, 2024), Gemini-2.0-Flash (Gemini Team et al., 2023). (2) open-source general-purpose MLLM Qwen2.5-VL-7B (Bai et al., 2025); and 3) specialized open-source reasoning MLLMs VLAA-Thinker-7B (Chen et al., 2025a), OpenVLThinker-7B (Deng et al., 2025), MMR1-Math-v0 (Leng et al., 2025), MM-Eureka (Meng et al., 2025a), and VL-Rethinker-7B (Wang et al., 2025).

## 5.2 Implementation Details

Our ReVisual-R1 models are based on the Qwen-2.5-VL-7B-Instruct and Qwen-2.5-VL-3B-Instruct models. Its training comprised three distinct stages. The process begins with a cold-start phase utilizing LLaMA Factory (Zheng et al., 2024) and pure text data to establish foundational language understanding. Following this, Multimodal Reinforcement Learning (MRL) is implemented using Easy R1 (Zheng et al., 2025). In this stage, the GRPO Kullback-Leibler (KL) divergence constraint is omitted to encourage broader policy exploration. The final stage TRL is also conducted via Easy R1. During TRL, the vision tower is frozen to concentrate learning on textual reasoning, and a small KL penalty is incorporated alongside entropy annealing to enhance training stability. All experiments are conducted on a setup of 8 NVIDIA A100-80G GPUs. Detailed prompt settings and training hyperparameters are provided in the Appendix.

## 5.3 Training Datasets

The training of ReVisual-R1 follows our proposed three-stage methodology, utilizing carefully curated datasets for each phase. The cold-start phase employed approximately 283k pure text entries focused on establishing foundational language understanding. Subsequently, the Multimodal Reinforcement Learning (MRL) phase used approximately 26k diverse multimodal entries from our GRAMMAR dataset to develop multimodal reasoning. Finally, the TRL phase consists of approximately 30k text entries designed to refine nuanced understanding and generation capabilities.

## 5.4 Main Results

As shown in Table 2, our model demonstrates superior performance on math-related benchmarks compared to other open-source reasoning models and even outperforms some commercial MLLMs.

Specifically, on the 7B model scale, ReVisual-R1-7B achieves an impressive average score of 53.1%, a significant improvement of +16.8 percentage points over the baselines in Tabel 2 on textual and multimodal benchmarks. Notably, ReVisual-R1 secures the top position among open-source contenders in nine out of ten individual benchmarks: MathVerse (+5.4% $\Delta$), MathVision (+18.6% $\Delta$), DynaMath (+4.5% $\Delta$), WeMath (+0.2% $\Delta$), LogicVista (+1.5% $\Delta$), AIME24 (+43.3% $\Delta$), AIME25 (+15.4% $\Delta$), GPQA (+10.1% $\Delta$), and MATH500 (+22.0% $\Delta$). All these results present the superior performance of our Revisual-R1 on both multimodal benchmarks as well as textual benchmarks.

When compared to closed-source commercial models, ReVisual-R1 also exhibits highly competitive performance. For instance, its average score (53.1%) surpasses that of OpenAI-GPT-4o (41.6%). On specific demanding benchmarks such as MATH500, ReVisual-R1 (89.2%) outperforms both doubao-1.5-vision-pro-32k (85.2%) and OpenAI-GPT-4o (74.6%). Similarly, on AIME24 and AIME25, ReVisual-R1 demonstrates substantial leads over these commercial offerings. While some closed-source models like doubao-1.5-vision-pro-32k show a higher overall average (55.8%), ReVisual-R1's ability to outperform them on several key reasoning tasks highlights its specialized strengths.

To further demonstrate the effectiveness of our training framework, we conduct experiments on 3B model scale, leading to Revisual-R1-3B. As demonstrated in Table 2, our model also demonstrates superior performance on this model size. Specifically, our model outperforms VLAA-Thinker-3B across all benchmarks and obtains an average 16.0% improvement. These results further verify the generalization of our Revisual-R1.

**Table 2:** Performance comparison on diverse benchmarks. The best scores are **bold**; the second best are underlined (among open-source models). Scores in *italics* indicate that they are not reported in the original work and are obtained using the VLMEvalKit (Duan et al., 2024) for evaluation. AIME24 and AIME25 results are averaged over eight independent inference runs to reduce score variance. **MathVerse-V**, **DynaMath-W** and **WeMath-S** denotes the vision-only, worst, and strict settings, respectively. Δ (e.g., **Ours-Open 3B Best**) denotes the improvement margin of the corresponding **ReVisual-R1** model over the best-performing **open-source** baseline model in the same scale across the respective column.

| | Multimodal Reasoning Benchmarks | | | | | | Textual Reasoning Benchmarks | | | | |
|---|---|---|---|---|---|---|---|---|---|---|---|
| Model | MathVerse-V | MathVision | MathVista | DynaMath-W | WeMath-S | LogicVista | AIME24 | AIME25 | GPQA | MATH500 | Avg. |
| *ClosedSource* | | | | | | | | | | | |
| doubao-1.5-vision-pro-32k | 64.7 | 51.5 | 78.6 | 44.9 | 64.2 | 65.7 | 26.7 | 20.0 | 56.1 | 85.2 | 55.8 |
| OpenAIGPT4o | 40.6 | 31.1 | 59.9 | 34.5 | 42.9 | 64.4 | 9.3 | 8.3 | 49.9 | 74.6 | 41.6 |
| Claude3.7Sonnet | 52.0 | 41.3 | 66.8 | 39.7 | 58.2 | 49.3 | 20.0 | 13.3 | 61.1 | 80.4 | 48.2 |
| Gemini2.0Flash | 43.6 | 47.8 | 70.4 | 42.1 | 47.4 | 52.3 | 33.3 | 36.7 | 35.4 | 69.0 | 47.8 |
| *3B-scale MLLMs* | | | | | | | | | | | |
| Qwen2.5-VL-3B | *33.0* | *22.7* | *60.0* | *9.0* | *17.9* | *28.8* | *6.7* | *0.0* | *22.2* | *57.2* | 25.8 |
| FAST-3B | *44.0* | *26.8* | *63.8* | *15.4* | *23.3* | *35.4* | *6.7* | *3.3* | *25.3* | *55.4* | 29.9 |
| VLAA-Thinker-3B | 36.4 | 24.4 | 61.0 | *18.2* | *33.8* | *38.5* | *3.3* | *0.0* | *27.8* | *61.4* | 30.5 |
| **ReVisual-R1-3B** | **46.2** | **46.0** | **64.8** | **24.6** | **39.6** | **45.4** | **40.0** | **36.7** | **37.4** | **84.2** | **46.5** |
| Δ (Ours-Open 3B Best) | +2.2 | +19.2 | +1.0 | +6.4 | +5.8 | +6.9 | +33.3 | +33.4 | +9.6 | +22.8 | +16.0 |
| *7B-scale Models* | | | | | | | | | | | |
| Qwen2.5VL7B | *38.7* | *26.6* | *68.2* | *12.6* | *24.5* | *35.6* | *10.0* | *6.7* | *32.8* | *67.2* | 32.3 |
| OpenVLThinker7B | *38.1* | *25.9* | *72.3* | *16.8* | *35.2* | *44.5* | *5.0* | *1.7* | *28.3* | *51.0* | 31.9 |
| MM-Eureka-Qwen-7B | *45.4* | *26.9* | *73.0* | *23.0* | *21.8* | *46.3* | *6.7* | *3.3* | *34.3* | *66.6* | 34.7 |
| MMR1-Math-v0 | 45.1 | *30.2* | *71.0* | *17.4* | *30.2* | *50.8* | *5.4* | *0.8* | *19.2* | *65.8* | 33.6 |
| ThinkLite7BVL | 42.9 | 24.6 | 71.6 | 16.5 | *41.8* | 42.7 | *8.8* | *27.9* | *24.8* | *61.4* | 36.3 |
| VLAA-Thinker-7B | *48.2* | 26.4 | 68.0 | 22.4 | 41.5 | 48.5 | *0.8* | *12.6* | *30.8* | *30.8* | 33.0 |
| VL-Rethinker-7B | 46.4 | 28.4 | **73.7** | 17.8 | 36.3 | 42.7 | *2.9* | *2.9* | *37.4* | *47.0* | 33.6 |
| **ReVisual-R1-7B** | **53.6** | **48.8** | 73.1 | **27.5** | **42.0** | **52.3** | **53.3** | **43.3** | **47.5** | **89.2** | **53.1** |
| Δ (Ours-Open 7B Best) | +5.4 | +18.6 | -0.6 | +4.5 | +0.2 | +1.5 | +43.3 | +15.4 | +10.1 | +22.0 | +16.8 |

**Table 3:** Ablation study of different training stage combinations applied to the ReVisual-R1 model, building upon a Cold Start. Best results per column are **bold** and second-best are underlined. Mixed-RL denotes that the model is jointly optimized with both MRL and TRL objectives in a mixed training stage.

| Training Strategies | MathVerse-V | MathVision | MathVista | DynaMath-W | WeMath-S | LogicVista | Avg |
|---|---|---|---|---|---|---|---|
| Cold Start (CS) only | 51.9 | 47.9 | 70.5 | 26.5 | 35.8 | 50.1 | 47.1 |
| CS + MRL | 50.9 | 47.6 | 71.9 | 25.7 | 38.8 | 51.2 | 47.7 |
| CS + TRL | 47.3 | 47.3 | 71.0 | 25.2 | 33.7 | 44.7 | 44.9 |
| **CS + MRL + TRL** | **53.6** | **48.8** | **73.1** | **27.5** | **42.0** | **52.3** | **49.6** |
| CS + TRL + MRL | 47.5 | 48.0 | 70.3 | 24.2 | 35.0 | 48.2 | 45.5 |
| CS + Mixed-RL | 49.3 | 48.2 | 72.1 | 25.7 | 38.8 | 51.2 | 47.6 |

Collectively, these results validate the efficacy of our proposed training method, including the structured three-stage curriculum and enhancements like Prioritized Advantage Distillation.

## 5.5 ABLATION STUDY

### 5.5.1 ABLATION ON SRO

To validate the effectiveness of our Staged Reinforcement Optimization (SRO) framework, we conduct ablation studies on different combinations of Multimodal RL (MRL) and Textual RL (TRL) phases, all building upon our optimized text-centric cold-start (CS). As shown in Table 3, the empirical results verify that our proposed CS + MRL + TRL (ReVisual-R1-MTR) sequence consistently yields the highest average performance (49.6 Avg). This outcome affirms our core hypothesis: an initial MRL phase establishes strong visual grounding, followed by a TRL phase to refine textual fluency and abstract reasoning, is crucial for developing superior multimodal capabilities.

**Table 4:** Ablation results demonstrating the impact of Prioritized Advantage Distillation (PAD) and its core components. Best results per column are **bold** and second-best are underlined.

| Model Configuration | Strategy | MathVerse | MathVision | MathVista | DynaMath | WeMath | LogicVista | Avg |
|---|---|---|---|---|---|---|---|---|
| ReVisual-R1 (CS + MRL) | PAD | **50.9** | **47.6** | **71.9** | 25.7 | **38.8** | **51.2** | **47.7** |
| *w/o PAD Components:* | | | | | | | | |
|   - Full PAD (Baseline) | GRPO-Baseline | 47.6 | 45.8 | 68.8 | 25.2 | 34.8 | 48.6 | 45.1 |
|   - No Prioritized Sub-sampling | GRPO-Filter | 47.7 | 46.7 | 71.2 | 25.5 | 35.1 | 49.7 | 46.0 |
|   - No Effective Sample Filtering | Random-Sampling | 47.9 | 46.4 | 70.7 | **26.1** | 37.1 | 49.3 | 46.2 |
| Other Strategy | DAPO | 48.3 | 46.3 | 69.2 | 25.4 | 38.3 | 49.2 | 46.1 |

In a more detailed analysis, the CS + MRL only model (47.7 Avg), while performing well on visually intensive tasks such as MathVista (71.9), does not reach the overall performance of the full MTR sequence. This further highlights the importance of the subsequent TRL stage. The alternative SRO ordering, CS + TRL + MRL (45.5 Avg), also proved less effective than our MTR approach. This finding indicates that establishing strong visual grounding before intensive textual refinement allows for more synergistic learning.

In addition, we perform a mixed-RL strategy to investigate the necessity of our staged reinforcement learning. In this setting, we train the cold-start model on a mixture of multimodal and textual data. However, as shown in Table 3, we observe that this mixed setting can only get an average 47.6 score, which is significantly lower than the performance of our staged RL. It shows that our MRL-then-TRL ordering within our SRO framework is a more effective strategy than simultaneously training on mixed-modality data in the RLVR stage.

### 5.5.2 ABLATION STUDY ON PAD

In this section, we conduct ablation studies to verify the effectiveness of our proposed Prioritized Advantage Distillation (PAD), examining its overall efficacy, the contribution of its components, and its sensitivity to key hyperparameters.

To assess PAD's impact, its full implementation is compared against GRPO-Baseline, GRPO-Filter-only, and Random-Sampling strategies. Table 4 demonstrates that full PAD achieved superior performance on mathematical reasoning benchmarks, highlighting the importance of its core components: effective sample filtering and prioritized sub-sampling. Meanwhile, we compare our method with DAPO and observe that our PAD can also present significantly better performance than DAPO. To further demonstrate the effectiveness of our Training dynamics (Figure **??**) further corroborate PAD's effectiveness, with its sampling strategy yielding higher reward accuracy and faster convergence.

### 5.5.3 GENERALIZATION OF REVISUAL-R1

To further evaluate the generalization capacity of our method, we test the model on several knowledge-intensive benchmarks, including MMMU, MMMU-Pro, and CMMMU, as well as on general-purpose multimodal perception (MMStar) and textual understanding (MMLU). The results, presented in Figure 5, show that our method consistently performs superior across these benchmarks. The more detailed performance is shown in Table 6. It further indicates that the generality of our approach on general tasks and provides further evidence that textual reasoning capabilities can benefit on general multimodal and textual tasks.

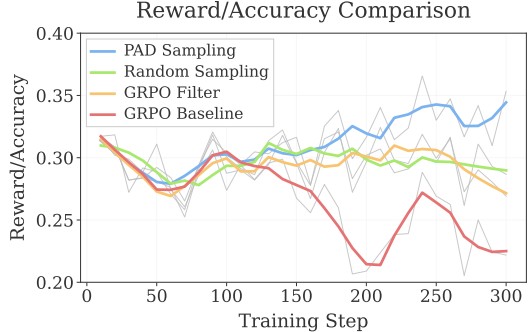

**Figure 4:** Ablation of training dynamics of our PAD.

## 6 RELATED WORK

### 6.1 MULTIMODAL LARGE LANGUAGE MODEL

Multimodal Large Language Model (MLLM) is a key research area. While leading closed-source models (e.g., GPT-o3 (OpenAI, 2025), Kimi-VL (Kimi Team, 2025)) excel at long Chain-of-Thought (CoT) reasoning, open-source contributions have focused on CoT adaptations (Liu et al., 2023; Guo et al., 2024; Su et al., 2024) and Supervised Fine-Tuning (SFT) with reasoning traces (Mitra et al., 2024b; Zhang et al., 2024c) (Gao et al., 2024). DeepSeek-R1 (Guo et al., 2025a) has further spurred Reinforcement Learning (RL) applications for visual reasoning (Huang et al., 2025b) (Dong et al., 2024) and specialized domains like mathematical reasoning. Nevertheless, many MLLM reasoning models (Wang et al., 2025) (Deng et al., 2025) (Yang et al., 2025) (Chen et al., 2025b) (Huang et al., 2025a) are limited by generating relatively short responses, which often curtail genuine reflection, thorough visual exploration, and consequently, deep multimodal reasoning. Our work, in contrast, introduces a novel framework to enable MLLMs to generate significantly longer, reflective responses, thereby facilitating long CoT reasoning to unlock more comprehensive multimodal reasoning capabilities.

### 6.2 REINFORCEMENT LEARNING IN REASONING

Current LLM research explores direct RL fine-tuning, specialized cold-start datasets for long-form reasoning, and advanced algorithms like Group Relative Policy Optimization (GRPO) (Shao et al., 2024) and its refinements (e.g., DAPO (Yu et al., 2025), DR.GRPO (Liu et al., 2025), GPG (Chu et al., 2025), AEPO (Chen et al., 2025c)) to elicit deeper reasoning. However, RL application to multimodal reasoning in MLLMs is nascent. Initial MLLMs efforts focus on subdomains like math reasoning (Meng et al., 2025b; Peng et al., 2025) or generative reward models (Gao et al., 2025), often utilizing data from commercial models. Nonetheless, successes such as DeepSeek-R1's (Guo et al., 2025a) rule-based RL are spurring similar MLLMs investigations, indicating growing interest in RL for unlocking sophisticated multimodal reasoning.

### 6.3 LIMITATIONS AND FUTURE WORK

Despite strong empirical results, ReVisual-R1 has several limitations. First, it lacks a solid theoretical foundation explaining why text-centric optimization strategiessuch as optimized cold start and staged multimodal/text-only RLeffectively improve reasoning. Second, scalability has only been validated on mid-sized models (3B and 7B), leaving uncertainty about performance on larger architectures like MoE models or other model families. Finally, the interaction between data type and training stage remains underexplored. Future work will systematically study the ratio and quality of multimodal versus textual data during cold start and RL phases to develop an optimal curriculum that maximizes performance across STEM reasoning, perceptual grounding, and general tasks.

## 7 CONCLUSION

This paper introduces ReVisual-R1, a 3B and 7B open-source MLLM designed to address prevalent challenges in cultivating sophisticated multimodal reasoning. By systematically integrating a strategic, high-difficulty text-only cold-start phase for foundational reasoning, a Multimodal RL stage employing GRPO stabilized by our novel Prioritized Advantage Distillation (PAD) mechanism, and a final TextRL refinement phase, our structured three-stage curriculum demonstrates that thoughtful data strategy and targeted algorithmic optimizations are pivotal. ReVisual-R1 achieves superior performance among open-source 7B models on a suite of challenging multimodal, textual reasoning and generalization benchmarks. This work underscores that careful curriculum design and algorithmic enhancements, rather than sheer model scale, can unlock robust, self-reflective multimodal reasoning.

## REPRODUCIBILITY STATEMENT

To ensure the reproducibility of our work, we provide an anonymous code repository in the supplementary materials containing the full implementation of our Revisual-R1 algorithm. All datasets used for training and evaluation are publicly available, and a comprehensive breakdown of our experimental setup, including all key hyperparameters, is detailed in Appendix.

## ETHICS STATEMENT

The development of our model, ReVisual-R1, and the curation of the GRAMMAR dataset were conducted with close attention to ethical guidelines. Our research is grounded in the use of publicly available and open-source academic datasets, which were rigorously filtered during the creation of GRAMMAR to ensure no personal, private, or sensitive information was included. The work does not involve human subjects, and its objective is to advance research in multimodal reasoning for the open-source community. We foresee no direct societal risks or harmful applications.

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

**Table 5:** Key Hyperparameters for Training Stages.

| Component | Hyperparameter | Component | Hyperparameter |
|---|---|---|---|
| **Cold Start** | Learning Rate = $2.0 \times 10^{-5}$
Gradient Accumulation = 8
Number of Epochs = 5
LR Scheduler = Cosine
Warmup Ratio = 0.05
Max Sequence Length = 32768
Precision = BF16
DeepSpeed = Zero2 | **Actor** | Global Batch Size = 128
Micro Batch Rollout = 4
Max Grad Norm = 1.0
Learning Rate (lr) = $1 \times 10^{-6}$
Weight Decay = $1 \times 10^{-2}$
Entropy Coef Init ($\beta_0$) = 0.02
Entropy Coef Min ($\beta_{\min}$) = 0.0
Entropy Decay Rate ($\lambda$) = 0.985 (exp)
Entropy Warmup Steps ($\tau_w$) = 140
Total Updates = 200000 |
| **GRPO** | Adv Estimator = grpo
KL Penalty Type = low var kl
KL Coef = $2 \times 10^{-3}$
$\tau = 0.3$ | **Model Settings** | Max Prompt Length = 8192
Max Response Length = 8192
Rollout Batch Size = 512
Generation Temperature = 1.0
Generation Top P = 0.95 |

# APPENDIX

APPENDIX CONTENTS

# A    LLM USAGE STATEMENT

In this paper, we employed a Large Language Model (LLM) in an assistive capacity for this manuscript. Its use was strictly limited for grammatical accuracy and refining sentence structure to improve clarity and readability. All intellectual contributions, including the formulation of ideas, data analysis, and the composition of the manuscript, are entirely the work of the authors.

# B    TRAINING SETTINGS

The training process can be divided into three distinct phases: cold start, multimodal reinforcement learning, and text-only reinforcement learning. Key hyperparameters for each training phase are detailed in Table 5.

# C    ALGORITHM IN PRIORITIZED ADVANTAGE DISTILLATION (PAD)

The PAD mechanism, introduced conceptually in the main text, is detailed in Algorithm 1 to clarify its step-by-step operation in refining training batches for more effective learning.

Initially, PAD filters the original batch $\mathcal{B}$ to create an "effective set" $\mathcal{E}$ of sample indices and a corresponding map $\hat{A}_{\mathcal{E}}$ for their advantages (Lines 2-10 in Algorithm 1). For each sequence $i$ in $\mathcal{B}$, its absolute advantage $|\tilde{A}_i|$ is computed. If this value falls within a specified informative range $[T_{low}, T_{high}]$, where $T_{low} > 0$ is crucial for excluding stagnant (near-zero advantage) samples, the index $i$ is added to $\mathcal{E}$, and its absolute advantage $\hat{A}_{i,abs}$ is stored in $\hat{A}_{\mathcal{E}}$.

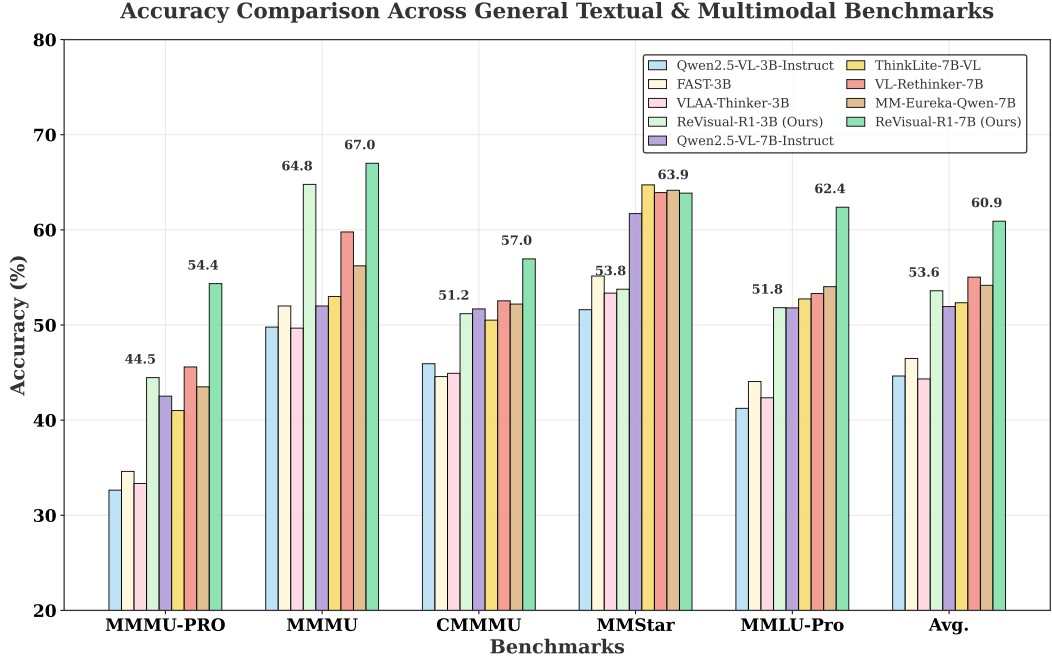

**Figure 5:** Performance comparison on general textual and multimodal benchmarks.

**Table 6:** Performance comparison on general textual and multimodal benchmarks. Best results per group (3B or 7B) are **bold**, and second-best per group (3B or 7B) are underlined.

| Model | MMMU | MMMU-PRO | CMMMU | MMStar | MMLU-Pro | Avg. |
|---|---|---|---|---|---|---|
| *3B Models* | | | | | | |
| Qwen2.5-VL-3B-Instruct | 49.78 | 32.64 | 45.93 | 51.61 | 41.24 | 44.64 |
| FAST-3B | 52.00 | 34.61 | 44.58 | **55.15** | 44.06 | 46.48 |
| VLAA-Thinker-3B | 49.67 | 33.34 | 44.92 | 53.35 | 42.35 | 44.33 |
| **Revisual-R1-3B** | **64.78** | **44.47** | **51.19** | 53.76 | **51.82** | **53.60** |
| *7B Models* | | | | | | |
| Qwen2.5-VL-7B-Instruct | 52.00 | 42.52 | 51.69 | 61.71 | 51.80 | 51.94 |
| ThinkLite-7B-VL | 53.00 | 41.01 | 50.51 | **64.73** | 52.74 | 52.34 |
| VL-Rethinker-7B | 59.78 | 45.59 | 52.54 | 63.92 | 53.31 | 55.03 |
| MM-Eureka-Qwen-7B | 56.22 | 43.50 | 52.20 | 64.16 | 54.03 | 54.17 |
| **Revisual-R1-7B** | **67.00** | **54.35** | **56.95** | 63.86 | **62.38** | **60.91** |

If this effective set $\mathcal{E}$ is non-empty, prioritized sub-sampling is performed (Lines 12-29). This multi-step process involves: (a) Calculating sampling probabilities $P_j$ for each sequence index $j \in \mathcal{E}$ via a temperature-controlled Softmax distribution over their stored absolute advantages $\hat{A}_{\mathcal{E}}[j]$ (Lines 14-21). A uniform probability distribution across $\mathcal{E}$ serves as a fallback mechanism should the Softmax normalization term $Z$ be zero. (b) Determining the sub-sample size $k'$ as $\min(\lceil \rho N \rceil, |\mathcal{E}|)$, where $\rho$ is the sub-sampling ratio and $N$ is the original batch size (Line 23). (c) Sampling $k'$ indices from $\mathcal{E}$ according to the calculated probabilities $P_{\text{dist}}$ (Line 25). (d) Constructing the final distilled mini-batch $\mathcal{B}_{\text{distilled}}$ by retrieving the original sequences corresponding to these $k'$ sampled indices (Lines 27-29). If $\mathcal{E}$ is void, an empty batch is returned. This entire procedure ensures that training batches are enriched by systematically filtering out uninformative data and prioritizing samples anticipated to yield more substantial learning signals.

## D    REASONING EXAMPLE

---

**Algorithm 1** Prioritized Advantage Distillation (PAD)

---

**Require:** Original batch $\mathcal{B} = \{seq_1, \ldots, seq_N\}$ with $N = |\mathcal{B}|$; advantage estimates $A_{\text{est}} = \{\tilde{A}_1, \ldots, \tilde{A}_N\}$; thresholds $T_{\text{low}}$, $T_{\text{high}}$; temperature $\tau$; subsampling ratio $\rho$
**Ensure:** Distilled minibatch $\mathcal{B}_{\text{distilled}}$
                     ▷ **Steps 1 & 2: Per-Sequence Advantage Calculation and Effective Sample Filtering**
 1: $\mathcal{E} \leftarrow \emptyset$                            ▷ Set of indices for effective samples
 2: $\hat{A}_{\mathcal{E}} \leftarrow \{\}$                ▷ Map: original index $i \in \mathcal{E} \rightarrow$ its absolute advantage $\hat{A}_i$
 3: **for** $i \leftarrow 1$ **to** $N$ **do**
 4:      $\hat{A}_{i,abs} \leftarrow |\tilde{A}_i|$                  ▷ Absolute advantage of current sequence $i$
 5:      **if** $T_{\text{low}} \leq \hat{A}_{i,abs} \leq T_{\text{high}}$ **then**
 6:          $\mathcal{E} \leftarrow \mathcal{E} \cup \{i\}$                  ▷ Add index to effective set
 7:          $\hat{A}_{\mathcal{E}}[i] \leftarrow \hat{A}_{i,abs}$         ▷ Store absolute advantage for effective sample $i$
 8:      **end if**
 9: **end for**
10: $\mathcal{B}_{\text{distilled}} \leftarrow \emptyset$
11: **if** $|\mathcal{E}| > 0$ **then**

                   ▷ **Step 3: Prioritized Sub-sampling from the Effective Set**
                   ▷ *a. Calculate sampling probabilities $P_j$ for each $j \in \mathcal{E}$*
12:      $Z \leftarrow \sum_{j \in \mathcal{E}} \exp(\hat{A}_{\mathcal{E}}[j]/\tau)$        ▷ Normalization term (Softmax denominator over $\mathcal{E}$)
13:      $P_{\text{dist}} \leftarrow \{\}$          ▷ Map: original index $j \in \mathcal{E} \rightarrow$ its sampling probability $P_j$
14:      **for all** $j \in \mathcal{E}$ **do**
15:          **if** $Z > 0$ **then**
16:              $P_{\text{dist}}[j] \leftarrow \exp(\hat{A}_{\mathcal{E}}[j]/\tau)/Z$
17:          **else**
18:              $P_{\text{dist}}[j] \leftarrow 1/|\mathcal{E}|$                ▷ Uniform fallback if $Z = 0$
19:          **end if**
20:      **end for**
                   ▷ *b. Determine actual sub-sample size $k'$*
21:      $k' \leftarrow \min(\lceil \rho N \rceil,\ |\mathcal{E}|)$
                   ▷ *c. Sample $k'$ indices from $\mathcal{E}$ according to probabilities $P_{\text{dist}}$*
22:      $S_{\text{sampled\_indices}} \leftarrow \text{SAMPLE}(\mathcal{E},\ P_{\text{dist}},\ k')$     ▷ $S_{\text{sampled\_indices}}$ is a list of $k'$ indices from $\mathcal{E}$
                   ▷ *d. Form the distilled mini-batch*
23:      **for all** $idx \in S_{\text{sampled\_indices}}$ **do**
24:          $\mathcal{B}_{\text{distilled}} \leftarrow \mathcal{B}_{\text{distilled}} \cup \{seq_{idx}\}$
25:      **end for**
26: **end if**
27: **return** $\mathcal{B}_{\text{distilled}}$

---

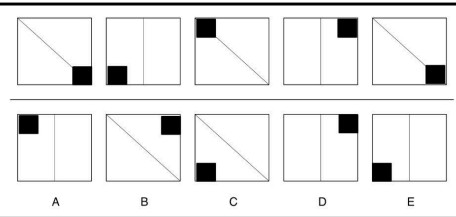

**Question:** 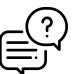

**Which of the boxes comes next in the sequence?**
**Select answers from A-E**

A  B  C  D  E

---

**Ours Revisual-R1 7B Model:**

<think>

Okay, let's try to figure out the sequence here. So there are five options labeled A to E, each with some black squares in different positions. The problem is asking which one comes next in the sequence.

**First, I need to look at the given figures and see if there's a pattern**. Let me list them out again to make sure I have them right, Top row:
1. A: A square divided by a diagonal, with a black square in the bottom right corner.
2. B: Two vertical stripes, each with a black square on the right side.
3. C: A square with a diagonal from top left to bottom right, and a black square in the top left quadrant.
4. D: Two vertical stripes, with a black square in the right stripe's upper part.
5. E: A square split into two vertical parts, with a black square in the left part.

**[Reflection]Wait**, maybe I should number them as per their order in the problem. The first row is options 1 to 5, then the second row is A to E. **Wait, the problem shows** the first row as the original five, then the next row labeled A to E.

**[Revisit Image]Wait, looking back**:The user presented the problem with two rows. The first row has five images (probably labeled 1 to 5), but the user then labels the second row as A to E. **Wait,** the actual question's options are A to E, each in the second row.

The sequence **seems to** involve the placement of the black squares relative to the diagonals or divisions of the square. **Let me analyze each option**:
Original top row (before the line):……
Then the next row is labeled A to E: A: Square split vertically, left third has a black square.……
**[Self-Critic]Wait, perhaps the original problem's figures are arranged as follows**:
First row (top): ……
Second row (labeled A to E): A: Vertical division, left part has a black square.……
**[Logic]Now, trying to see the pattern. Let's look at the original figures (before the line) and then the options A-E.** The task is to determine which comes next.Looking at the first row's figures (original positions):……
**[Self-Critic]Wait, maybe the pattern alternates between diagonal and vertical splits. Let's check**:
Original sequence (top row):1. Diagonal, lower right.……
Then the next row (A-E): A: Vertical split (three parts?), left part black.……
If the pattern is alternating between diagonal and vertical splits, then after figure 5 (diagonal), the next would be a vertical split.
**[Hypothesis]** Looking at the original vertical splits: figures 2, 4, and **maybe** A, D, E.
Let's see: Original vertical splits (two parts) are figures 2 (lower right), 4 (upper right). Then after figure 5 (diagonal), next could be another vertical split. Among options A-E, which are vertical splits: A: vertical split into three columns? **Or two columns?** The image for A is a square split vertically into left and right, with a black square in the left part. D is vertical split into two, with black in the right upper. E is vertical split into two, black in left lower.
So after figure 5 (diagonal), the next might be a vertical split. The last vertical split in the original sequence (figure 4) has the black square in the upper right. Then the next vertical split could be in the lower left (E) or upper left (A?), **but need to check the order**.
**[Hypothesis & Analysis]** Alternatively, looking at the positions:\n\nOriginal vertical splits:\n\nFigure 2: right lower.\n\nFigure 4: right upper.\n\nSo if the pattern for vertical splits is alternating between lower and upper in the right section, then next could be lower **right again?** **But** figure 2 is right lower, figure 4 is right upper. So **maybe** the next vertical split would be right lower again. Then option E is left lower. **Hmm, not sure**.
Alternatively, **maybe** the sequence is moving in a certain direction. Let's think of the figures as arranged in rows. The first row has five figures, and the second row has A-E. The correct answer is among A-E, which are the second row.
**[Hypothesis] Alternatively, the sequence might be based on rotation or the position of the black square**. For example, starting with diagonal, then vertical splits. Let's see:…… **Wait**, the options A-E are all vertical splits except B and C, which have diagonals. **Wait,** original figures 1-5 have a mix: 1,3,5 have diagonals; 2,4 have vertical splits. The options A-E: A is vertical split left; B is diagonal with black in upper right; C is diagonal upper left; D is vertical split right upper; E is vertical split left lower.
So after figure 5 (diagonal lower right), the next figure is likely a vertical split. Now, in the original vertical splits (figures 2 and 4), figure 2 is right lower, figure 4 is right upper. **So maybe the next vertical split alternates.** If figure 4 was right upper, then next could be right lower again (figure 2's position).
**[Reflection] But** option E is left lower. **Hmm. Alternatively, the vertical splits in the original sequence go from right lower to right upper.** If the next vertical split is right lower again, **but** that's figure 2. **But E is left lower**.
**[Hypothesis] Alternatively, maybe the direction is moving left**.
**Wait, maybe** the pattern is that after two diagonal figures (1,3,5), the vertical splits alternate between right and left. Let's see:\n\nOriginal vertical splits: figure 2 (right lower), figure 4 (right upper). Then the next vertical split might be left lower (E) or left upper (A). **But** figure A is left black square, which is a vertical split left. If the pattern is moving from right to left, then after figure 4 (right upper), next would be left lower (E).
**[Self-Critic] But I need to check**. Alternatively, looking at the positions of the black squares in the vertical splits:\n\nFigure 2: right third, lower part.\n\nFigure 4: right third, upper part.
So the vertical split's black square moves from lower to upper in the same right third. Then, if the next vertical split is in the left third, **maybe lower part (E)**. So after moving right upper, next is left lower. That would make the pattern: right lower, right upper, left lower. So E. Alternatively, the answer is E.
**[Self-Critic] Let's check the options:** Original sequence: diagonal, vertical split (right lower), diagonal, vertical split (right upper), diagonal. Then next is vertical split (left lower, E). That seems plausible.Another way: looking at the positions of the black squares in the vertical splits. The vertical splits in figures 2 and 4 have the black square in the right third. Figure 2's black square is at the lower part of the right third, figure 4's is at the upper part.
**So if we follow that pattern, the next vertical split would be in the left third, perhaps at the lower part (E). So the answer is E.\n**

</think>

<summary>

 The sequence involves alternating between diagonal and vertical split figures. After the diagonal figure (5), the next is a vertical split. The vertical splits in the original sequence (figures 2 and 4) have black squares in the right third, moving from lower to upper. The next vertical split follows this pattern but in the left third, at the lower part.
**Answer: \boxed{E}**  ✓

</summary>

---

**Figure 6:** Our Revisual-R1 model reasoning case, showcasing its exceptional reasoning ability. The model generates long responses, continuously hypothesizing, reflecting, verifying, and correcting to arrive at the final answer, while also providing a summary answer.

