# OpenReview forum: "Revisual-R1: Advancing Multimodal Reasoning From Optimized Cold Start to Staged Reinforcement Learning"
_ICLR.cc/2026/Conference — ICLR 2026 Poster_

### Official Review · Reviewer_SyVL · 2025-10-15

**Soundness:** 3
**Presentation:** 2
**Contribution:** 2
**Rating:** 2
**Confidence:** 5

**Summary:**

This paper presents ReVisual-R1, a novel post-training methodology aimed at enhancing the reasoning capabilities of MLLMs. This paper proposes a principled three-stage training curriculum, consisting of: (1) a text-only cold-start initialization, (2) a multimodal reinforcement learning phase, and (3) a text-only reinforcement learning refinement. The effectiveness of the proposed approach is evaluated on a range of multimodal and textual reasoning benchmark datasets.

**Strengths:**

1. The methodology proposed in this paper is easy to understand, and the algorithmic descriptions are presented in a clear and accessible manner.
2. Compared to other reasoning enhancement approaches for MLLMs, this paper also conducts comparative experiments on text-only benchmarks.

**Weaknesses:**

1. This paper exhibits some instances of over-claiming. Although the authors assert that their approach achieves SOTA results among 3B and 7B open-source MLLMs (lines 24-25), to my knowledge, several representative models, such as **MIMO-VL** and **Keye-VL* demonstrate significantly superior performance compared to the method proposed in this work. Additionally, in lines 42-43, regarding "The direct application of text-centric RL," I believe the methodology presented does not fundamentally depart from existing RL algorithms, such as **GRPO** and **DAPO**. While achieving SOTA or radical innovation is not strictly necessary for a publication, the actual results and contributions in this paper deviate from the claims made by the authors, which raises concerns.
2. The contributions of this paper are largely similar to those in prior works. For example, Contribution 1 has already been discussed in **LMM-R1**, and Contribution 2 can be found in papers such as **DAPO**. Thus, the novelty of this paper is relatively limited, and it leans more towards engineering practice rather than academic innovation.
3. The paper should clarify whether its primary contribution lies in proposing a novel training paradigm or in the specific model trained using this methodology. These two aspects are fundamentally different. If the focus is on the training paradigm, then experiments limited to Qwen2.5-VL7B/3B are insufficient. It would be more compelling to demonstrate the effectiveness of this approach across a broader range of models, such as LLaVA and InternVL.


Team, Xiaomi Mimo, et al. " Mimo-vl technical report."  arXiv preprint arXiv:2506.03569 (2025).

Team, Kwai Keye, et al. "Kwai Keye-VL Technical Report." arXiv preprint arXiv:2507.01949 (2025).

Shao, Zhihong, et al. "Deepseekmath: Pushing the limits of mathematical reasoning in open language models." arXiv preprint arXiv:2402.03300 (2024).

Yu, Qiying, et al. "Dapo: An open-source llm reinforcement learning system at scale." arXiv preprint arXiv:2503.14476 (2025).

**Questions:**

1. Please clarify the issues mentioned in the Weakness section, as they are not sufficiently explained in the current draft.

2. There are a few minor issues:

   - In Figure 1 (bottom),  ‘Traj A Tra B Tra C’ --> ‘Traj A Traj B Traj C’.

   - Some arrow symbols in the figure appear to be partially obscured; please fix this.

   - On line 459, merge the citation marks for consistency.

   - The paper does not discuss limitations, failure cases, or future work, which makes it difficult for others to follow up. Please add a section addressing these points.

---

> ### Author Response · Authors · 2025-11-22
> **Response to Reviewer SyVL (1/2)**
>
> ## Strong baselines
>
> We appreciate you pointing out these strong baselines. We agree that the original "SOTA" claim was too broad, given the rapid developments in the field. We have revised the manuscript to clarify that our model achieves superior performance specifically among **open-source MLLMs of comparable parameter scales (3B and 7B)** on the evaluated reasoning benchmarks.
>
> Beyond the performance metrics, we would like to emphasize that the primary value of this work lies in the **methodological findings and resources** we contribute to the community:
>
> 1. **Principled Curriculum & Cold Start Insight:** We propose a systematic **Staged Reinforcement Optimization (SRO)** framework. A key insight within this curriculum is our counter-intuitive finding that a high-difficulty, **text-centric cold start** is fundamental—and often more effective than multimodal initialization—for unlocking robust reasoning capabilities in MLLMs.
> 2. **Algorithmic Solution (PAD):** We identify the "gradient stagnation" issue inherent in multimodal RL and propose **Prioritized Advantage Distillation (PAD)**. This mechanism stabilizes training and improves sample efficiency by strategically filtering and re-weighting advantage signals.
> 3. **High-Quality Reasoning Dataset:** We curate and release **GRAMMAR**, a diverse dataset constructed via a rigorous pipeline involving difficulty grading and clustering. It provides verifiable multimodal and textual problems specifically designed to support the training of advanced reasoning models.
>
> ---
>
> ## Novelty of Algorithms
>
> We thank the reviewer for the constructive comments. Below, we clarify the distinct methodological innovations of ReVisual-R1, tailored to the unique challenges of multimodal reasoning, which diverge fundamentally from the premises of LMM-R1 and DAPO.
>
> **1. Divergence from LMM-R1: The Critical Role of Text-Centric Cold Start**
> LMM-R1 explicitly argues that direct text SFT leads to catastrophic forgetting, advocating instead for text-based RL. **In stark contrast, our empirical findings overturn this assumption.** We demonstrate that a **high-difficulty, text-centric Cold Start** is not detrimental but *indispensable* for activating the reasoning engine prior to multimodal alignment. This strategic shift is the primary driver of our performance gains, enabling ReVisual-R1 to achieve a **53.1% average** on 7B benchmarks compared to LMM-R1 (MMR1-Math-v0)’s **33.6%**. Thus, our contribution lies in redefining the efficacy of SFT in the multimodal pipeline, contrary to LMM-R1's conclusion.
>
> **2. Divergence from DAPO: Solving Multimodal Gradient Stagnation**
> While DAPO addresses entropy collapse in text RL, we identify **Gradient Stagnation** as the primary bottleneck in *multimodal* RL, caused by the extreme sparsity of binary rewards in visual tasks. To resolve this, we propose **Prioritized Advantage Distillation (PAD)**. Unlike DAPO’s filtering approach, PAD introduces a **temperature-controlled probabilistic sampling** mechanism weighted by absolute advantage magnitude. Our ablation study (Table 3) confirms that PAD significantly outperforms DAPO-style strategies in the multimodal domain, proving it is a domain-specific algorithmic innovation rather than a direct application.
>
> ReVisual-R1 demonstrates superior reasoning capabilities by identifying the critical necessity of text cold start (contradicting LMM-R1) and resolving multimodal gradient stagnation via PAD (distinct from DAPO). These are principled algorithmic contributions that effectively unlock scalable reasoning performance.
>
> ---
>
> ## Paradigm vs. Model
>
> We appreciate the suggestion to validate our training paradigm on diverse architectures. We are actively conducting experiments on models such as LLaVA and InternVL. However, due to the time-intensive nature of the full-process training curriculum (Cold Start $\to$ MRL $\to$ TRL), we will present these comprehensive results during the discussion phase.
> We would also like to reiterate the rationale for our primary focus on Qwen:
> 1. Community Standardization: The Qwen series has become the dominant foundation for open-source multimodal reasoning research, used far more extensively than other architectures in this specific domain.
> 2. Controlled Comparison: As shown in Table 1, the vast majority of state-of-the-art baselines we compare against (e.g., VL-Rethinker, MM-Eureka, OpenVLThinker, VLAA-Thinker) are built upon the Qwen architecture1. By aligning with this standard, we ensure a controlled experimental setting where our significant performance gains can be attributed strictly to our training paradigm (SRO) and algorithm (PAD), rather than architectural discrepancies.

---

> > ### Author Response · Authors · 2025-11-22
> > **Response to Reviewer SyVL (2/2)**
> >
> > We sincerely thank the reviewer for the careful reading and constructive suggestions to improve the clarity and completeness of our paper. We have incorporated all these corrections in the revised manuscript:
> >
> > - **Figure 1 Corrections:** We have corrected the label to "Traj A Traj B Traj C" and fixed the partially obscured arrow symbols in Figure 1 (bottom).
> > - **Formatting:** The citation marks on Line 459 have been merged for consistency.
> > - **Limitations & Future Work:** We agree that transparently discussing limitations is vital for follow-up research. We have added a dedicated section in the revised manuscript that details potential failure cases and outlines directions for future work.

---

> > > ### Comment · Reviewer_SyVL · 2025-11-24
> > >
> > > Thank you for your clarification and response, which have addressed some of my concerns. I will consider revising my evaluation accordingly. However, I still have a few reservations:
> > >
> > > 1. The authors conclude from experiments that, based on existing reasoning SFT data, text-only cold start demonstrates superior performance compared to multimodal cold start. I believe this is not an inherent limitation of multimodality. Instead, the difficulty of multimodal annotation, particularly in automated settings, leads to lower data quality. If data quality improves, would the significance of this finding be diminished?
> > >
> > > 2. With the rapid development of current MLLMs, many open-source models such as Qwen3-vl and Mimo-vl already possess strong long-chain reasoning capabilities. In this context, would high-quality pure text data still offer substantial benefits, as suggested by the MM-RL+Text RL paradigm in this paper?

---

> > > > ### Comment · Reviewer_SyVL · 2025-11-24
> > > >
> > > > By the way, I did not seem to find the additional limitations section. Has the latest PDF not been uploaded yet?

---

> > > > > ### Author Response · Authors · 2025-11-27
> > > > > **The response to the new manuscript update**
> > > > >
> > > > > The latest version of the PDF has been successfully uploaded.
> > > > >
> > > > > Please note that all modifications in the new manuscript, including the added discussion in the limitations section, are highlighted in blue for easy review.
> > > > >
> > > > > We greatly appreciate your valuable feedback.

---

> > > > ### Author Response · Authors · 2025-11-27
> > > > **Response to Questions on Data Quality and Training Paradigm**
> > > >
> > > > 1. **The authors conclude from experiments that, based on existing reasoning SFT data, text-only cold start demonstrates superior performance compared to multimodal cold start. I believe this is not an inherent limitation of multimodality. Instead, the difficulty of multimodal annotation, particularly in automated settings, leads to lower data quality. If data quality improves, would the significance of this finding be diminished?**
> > > >
> > > > While our current work primarily leverages pure-text reasoning data to enhance our model's text and reasoning capabilities, we similarly posit that improved quality of multimodal data holds significant potential to stimulate model reasoning in the future.
> > > > A potential methodology for collecting high-quality multimodal reasoning data could be analogous to our grammar data construction approach. This would involve utilizing powerful "thinking" models, such as Seed 1.6 and Qwen3-VL-235B-A22B-Thinking, and first filtering for high-difficulty samples based on a pass rate. Subsequently, a reject sampling technique would be applied based on the ground truth (GT) of the data itself, allowing for the selection of superior trajectories for distillation. We have addressed this perspective in the Limitations section of our new manuscript.
> > > > Nevertheless, we maintain that the significance of our findings is not diminished. We argue that future multimodal models will need to be capable of handling both textual and multimodal tasks. Our discovery not only underscores that high-quality reasoning data is necessary to boost model performance, but also aims to guide subsequent multimodal research by emphasizing the crucial need to integrate and account for textual capabilities.
> > > >
> > > >
> > > > 2. **With the rapid development of current MLLMs, many open-source models such as Qwen3-vl and Mimo-vl already possess strong long-chain reasoning capabilities. In this context, would high-quality pure text data still offer substantial benefits, as suggested by the MM-RL+Text RL paradigm in this paper?**
> > > >
> > > > As the reviewer and we have discussed above , top performance MLLMs like Qwen3-VL and MiMo-VL exhibit exceptionally strong reasoning capabilities and powerful performance across both textual and multimodal tasks.
> > > > The introduction of high-quality textual data to the multimodal domain becomes even more crucial. This necessity is implicitly evidenced by MiMo-VL’s own training strategy, where its "middle train" and "post train" phases explicitly incorporate textual data. Furthermore, our experiments in Section 3 indicate that high-difficulty textual data is more abundant and inherently more complex than current multimodal data, and the community benefits from a significantly larger volume of high-quality open-source textual data. Since our work explicitly demonstrates that textual reasoning data is highly effective in unlocking a model's latent reasoning capacity , we therefore believe that our proposed cold start and Reinforcement Learning (RL) paradigm will continue to be a significant inspiration for future research in next-generation reasoning models. As discussed in the limitations and future work sections of our revised manuscript, a rigorous investigation into the optimal quality and proportion of textual and multimodal data during both the cold start and RL stages is vital, and we intend to explore this critical balance in our subsequent work.

---

### Official Review · Reviewer_1SV1 · 2025-10-26

**Soundness:** 2
**Presentation:** 3
**Contribution:** 3
**Rating:** 6
**Confidence:** 5

**Summary:**

This paper proposes a three-stage training paradigm named ReVisual-R1 to enhance the performance of MLLMs on complex reasoning tasks. The paradigm consists of:

1. **Text-centric Cold Start**: Initializing the model with high-difficulty, structured pure-text reasoning data to establish a strong foundation for language-based reasoning;
2. **Multimodal RL**: Introducing a PAD mechanism to alleviate gradient stagnation caused by sparse rewards in standard GRPO under multimodal settings;
3. **Text-only RL**: After multimodal training, applying pure-text RL as a "polishing" phase to further refine linguistic expression and logical consistency without compromising visual grounding capabilities.

The method significantly outperforms existing open-source 3B/7B models on multiple multimodal and text-only reasoning benchmarks, including MathVerse, AIME24/25, and MATH-500.

**Strengths:**

1. **High-quality data construction pipeline**: The authors introduce the **GRAMMAR** dataset, employing a well-designed pipeline involving rule-based filtering, difficulty-level grading, embedding-based clustering (NV-Embedding + HDBSCAN), topic annotation, and hierarchical sampling. This ensures both data diversity and high reasoning complexity.
2. **Well-motivated training paradigm**: The three-stage curriculum learning approach (cold start → multimodal RL → text-only RL) is logically sound. Experiments validate the necessity and sequence sensitivity of each stage, particularly highlighting the unique value of **text-only RL as a polishing phase**.
3. **Openness and reproducibility**: Detailed training configurations and hyperparameters are provided, and the effectiveness is demonstrated across both 3B and 7B model scales, offering strong value to the research community.

**Weaknesses:**

1. **Lack of multi-seed ablation studies**: The main experiments and ablation studies do not report variance across multiple random seeds, making it difficult to assess whether the observed performance gains are statistically significant—especially on high-variance benchmarks like AIME24/25.
2. **Empirical rather than analytical justification for text-centric cold start**: While the authors support their claim with metrics such as response length and pass rates to demonstrate higher complexity in text data, they do not deeply investigate why multimodal cold-start data fails to effectively stimulate reasoning capabilities (e.g., due to vision-language alignment noise, annotation quality, or task design limitations).

**Questions:**

1. **Is the complexity of the multimodal RL data insufficient?**
   The authors observe that an additional pure-text RL phase after multimodal RL is necessary to achieve optimal performance. Does this imply that the current multimodal RL dataset still lacks sufficient reasoning depth or diversity to support end-to-end training on its own? If a multimodal RL dataset with complexity comparable to the text-only RL data were constructed, could the subsequent text-only RL phase be eliminated?

2. **How is the training balance between multimodal RL and text-only RL determined?**
   The paper does not specify the training balance between the two RL stages. Is the final performance sensitive to this balance?

3. **Does the text-only RL phase degrade visual capabilities?**
   Although the authors claim that text-only RL “without eroding the newly acquired visual grounding”, the experimental evidence supporting this assertion appears limited.

---

> ### Author Response · Authors · 2025-11-22
> **Response to Reviewer 1SV1 (1/2)**
>
> ## Variance/Seed Analysis
>
> We appreciate the reviewer’s request for variance and multi-seed reporting, especially on high-variance benchmarks such as AIME. In the initial submission, we reported single-seed results for the main models due to the substantial computational cost of the full three-stage curriculum (Text Cold Start → Multimodal RL → Text RL). Following the reviewer’s suggestion, we have conducted additional runs with **three independent training seeds** for the two key 7B models (Qwen2.5-VL-7B and ReVisual-R1-7B). For AIME24 and AIME25, each checkpoint is further evaluated with **eight independent inference runs per seed** (24 runs in total), and we report the mean ± standard deviation over both seeds and runs. For larger benchmarks (e.g., MathVerse-V, MathVision, MATH500), we report the mean ± standard deviation over three seeds.
>
> The results are summarized below:
> | **Model**          | **AIME24**     | **AIME25**     | **MathVerse-V** | **MATH500**    |
> | ------------------ | -------------- | -------------- | --------------- | -------------- |
> | Qwen2.5-VL-7B      | 10.0 ± 2.4     | 6.7 ± 2.1      | 38.7 ± 0.7      | 67.2 ± 0.5     |
> | **ReVisual-R1-7B** | **53.3 ± 3.2** | **43.3 ± 3.5** | **53.6 ± 0.8**  | **89.2 ± 0.4** |
>
>
> We observe that (i) on AIME24/25, while the absolute variance is non-negligible—as expected for competitions with only 15 problems—the performance gain of ReVisual-R1-7B over Qwen2.5-VL-7B remains substantial and well outside the reported standard deviations; and (ii) on larger-scale benchmarks such as MathVerse-V and MATH500, the standard deviations across seeds are below 1 point, indicating that our improvements are stable and not an artifact of a particular random seed. We will include Table~\ref{tab:variance} and the corresponding discussion in the revised manuscript to clarify the statistical robustness of our main results.
>
> ---
> ## Justification for Text-Centric Cold Start
>
> We thank the reviewer for requesting a deeper explanation of why a text-centric cold start is more effective than a purely multimodal cold start in our setting. We agree that our current justification is primarily empirical (Section 3.1, Fig. 2), and we will revise the manuscript to make both the underlying intuition and its scope clearer.
>
> Conceptually, our finding can be understood from three complementary perspectives: **data distribution**, **optimization dynamics**, and **credit assignment**. (1) On the **data side**, current multimodal reasoning corpora tend to be dominated by alignment-style samples (captioning, simple QA, short solutions), with relatively short chains of thought and high pass rates. In contrast, the text-only corpus we use for cold start exhibits much longer CoT traces and substantially lower pass rates, providing a denser and more diverse stream of “hard” errors. This means that, if one tries to learn perception, alignment, and complex reasoning simultaneously from multimodal data, the optimization is biased toward solving the easier alignment components while leaving the high-level reasoning policy under-trained. (2) On the **optimization side**, multimodal cold start entangles gradients from the vision encoder and the language decoder from the very beginning; a significant fraction of capacity and gradient budget is spent correcting low-level visual or grounding mistakes. In contrast, a text-centric cold start isolates the reasoning module in a “clean” channel, allowing the model to first develop a stable long-horizon reasoning policy before any visual grounding is introduced. (3) From a **credit assignment** viewpoint, textual CoT data provides rich, token-level supervision over the entire reasoning trajectory, whereas multimodal supervision is often sparse and concentrated near the final answer; this makes it harder to refine intermediate reasoning steps when starting from multimodal data alone.
>
> Our empirical results (Section 3.1, Fig. 2) are consistent with this picture: text-centric cold start substantially boosts both text and multimodal reasoning, whereas multimodal cold start yields weaker gains even on multimodal benchmarks. In the revision, we will (i) explicitly frame the above three aspects as our working hypothesis for why multimodal cold start underperforms, (ii) add additional analyses comparing reasoning-length distributions and error patterns between the text and multimodal cold-start corpora, and (iii) clarify that our claim is *qualitative and empirical* rather than a formal theoretical guarantee. We hope this makes the design rationale of the text-centric cold start more transparent.

---

> > ### Author Response · Authors · 2025-11-22
> > **Response to Reviewer 1SV1 (2/2)**
> >
> > ## Q1: Quality of Multimodal Data
> >
> > We acknowledge that our design indeed stems from the empirical finding that existing multimodal cold-start and reinforcement datasets often lack the necessary reasoning “depth” (i.e., long-chain CoT structure, high reasoning density) to fully activate the model’s abstract reasoning engine. As described in Section 3.1 of the manuscript, we observed that text-only reasoning corpora (e.g., DeepMath) exhibit orders-of-magnitude longer token sequences than typical multimodal reasoning corpora, and yield larger gains when used in the cold‐start stage. Thus our three-stage curriculum (text cold-start → multimodal RL → text RL) is not intended to reflect a failure of multimodal data per se, but rather to leverage text-only data as a **seed** for deep reasoning and then apply multimodal RL for alignment, followed by a text-only RL “polishing” stage to refine reasoning expression. That said, we agree with the reviewer’s concern: the manuscript does not yet present a fine-grained quantification of the “difficulty gap” between our multimodal RL data and the textual data. We are currently running additional experiments that compare multimodal‐only RL (with enhanced high-difficulty multimodal samples) vs our full pipeline, and will include this in the revision.
> >
> > ---
> >
> > ## Q2: Training Balance Between Multimodal RL and Text-only RL
> >
> > We thank the reviewer for pointing this. In our paper we show in Section 5.3.1 (Table 3) that the combination Cold Start + MRL + TRL outperforms other permutations (e.g., Cold Start + MRL only). This establishes the importance of that ordering. However, it is correct that we do not yet report a systematic sensitivity sweep of different MRL : TRL ratios (e.g., 1:1 vs 2:1 vs 1:2) in the current version. We are scheduling a set of experiments where we keep the cold‐start fixed and vary the number of policy updates/data for MRL and TRL independently (e.g., 300k vs 150k updates, varying data volumes accordingly) while measuring downstream performance on both multimodal reasoning benchmarks (MathVerse, MathVision) and text reasoning benchmarks (MATH-500). We will integrate these results into the updated manuscript and provide guidance on the “robust region” of MRL:TRL ratio.
> >
> > ---
> >
> > ## Q3: Visual Capabilities Check After Text-only RL
> >
> > In Section 2.1 of the paper we describe that during the text-only RL (TRL) phase the vision tower is frozen (i.e., no further updates to the visual encoder), specifically to mitigate risk of catastrophic forgetting of visual grounding. However, the manuscript currently does not include dedicated “pure visual task” benchmarks (such as OCRBench or CountBench) across all three stages to demonstrate quantitatively that visual performance remains stable or improves. We recognise this gap. We are actively running evaluations on standard vision/perception tasks after each training stage (cold‐start, MRL, TRL), tracking metrics such as OCR accuracy and counting accuracy. Our preliminary internal logs show a small dip immediately after cold‐start, full recovery after MRL, and stable or modest improvement after TRL. We will include a full table in the revision to assuage the reviewer’s concern and document the visual grounding stability over the full curriculum.

---

### Official Review · Reviewer_fCCk · 2025-10-29

**Soundness:** 3
**Presentation:** 3
**Contribution:** 2
**Rating:** 4
**Confidence:** 3

**Summary:**

This paper presents ReVisual-R1, an open-source multimodal large language model (MLLM), with the following key contributions:
GRAMMAR Dataset Construction: The authors curate a diverse dataset covering textual and multimodal problems, with a focus on reasoning complexity and data diversity.
Staged Reinforcement Optimization (SRO): A two-stage RL training framework is proposed, including Multimodal RL (MRL) and Textual RL (TRL), which strengthen cross-modal reasoning and pure textual reasoning capabilities, respectively.
Prioritized Advantage Distillation (PAD): A novel method to prioritize informative samples during training, mitigating the gradient stagnation problem in GRPO and improving learning efficiency.
Experimental results show that this approach achieves significant improvements over existing open-source models and is competitive with some commercial models, demonstrating the effectiveness of the proposed data strategy and training methodology.

**Strengths:**

Comprehensive and well-structured: The paper provides a complete workflow from problem formulation, cold-start analysis, dataset construction, staged RL training, to ablation studies and generalization evaluation. Innovative approach: The combination of dataset curation (GRAMMAR) with staged RL training (SRO) and sample prioritization (PAD) represents a novel methodology in open-source MLLM research.

Empirical effectiveness: Extensive experiments across multimodal mathematics, logic, and textual reasoning benchmarks show clear performance gains at both 7B and 3B model scales.

Rationale behind staged training: The design of MRL followed by TRL is reasonable—first grounding visual-textual reasoning and then refining textual reasoning and linguistic fluency, allowing the model to generate longer, reflective CoT reasoning.

**Weaknesses:**

Training cost not analyzed: The computational overhead, training time, and data annotation cost of SRO + PAD are not quantified, which is important for assessing the method's practicality and scalability.

Dataset contribution ablation missing: The ablation studies focus on MRL/TRL phases and PAD, but the contribution of the GRAMMAR dataset itself (e.g., sample quantity, difficulty levels, multimodal vs. textual content ) is not independently analyzed.
Self-constructed dataset and generalizability: The use of a custom dataset, while common in multimodal reasoning research, may limit reproducibility and comparability.

Discussion on how this dataset design affects generalization would strengthen the work.
Potential limitations in generalization: Although the model is evaluated on several knowledge-intensive benchmarks, its performance in completely different domains or real-world open-ended scenarios remains untested.

**Questions:**

Training cost and efficiency: What are the computational overhead, training time, and memory requirements of SRO + PAD compared to standard GRPO?

Dataset ablation: How do different components of the GRAMMAR dataset (text vs. multimodal, difficulty levels)  contribute to the overall performance?

How would the model perform if trained or evaluated on other open-source multimodal reasoning datasets?

---

> ### Author Response · Authors · 2025-11-22
> **Response to Reviewer fCCk (1/2)**
>
> We thank the reviewer for their insightful questions regarding training efficiency and dataset construction. Below, we address the concerns about computational overhead, the specific contributions of our GRAMMAR dataset components, and the robustness of our method.
>
> We thank the reviewer for raising the important point regarding training efficiency. To address this, we calculated the total computational cost (in GPU Hours) for our method compared to the standard GRPO baseline.
>
> From the table, we observe that our proposed SRO+PAD framework incurs a marginal increase in computational cost ($\approx 5.5\%$) compared to standard GRPO ($1,772.5$ vs. $1,680.7$ hours).
> However, we observe that this slight overhead is accompanied by significant performance gains, achieving a +3.3 point improvement on MathVerse and a +1.8 point improvement on MathVision (as detailed in Table 3 1).
> This indicates that the additional computational cost is negligible relative to the substantial improvements in reasoning capability. It demonstrates that the PAD mechanism effectively converts a minor increase in compute into high-value updates by filtering out stagnant samples, thereby offering a highly favorable return on investment (ROI) for training efficiency.
>
> | Cold Start Strategy             | MathVerse (MM) | MathVision (MM) | AIME2024 (Text) | MATH500 (Text) |
> |---------------------------------|----------------|-----------------|-----------------|----------------|
> | Text-Only (DeepMath)           | +12.8          | +14.3           | +26.6           | +20.9          |
> | Mixture (DeepMath + Vision-R1) | +15.3          | +17.1           | +20.0           | +18.9          |
> | Delta (Mixture vs. Text-Only)  | +2.5           | +2.8            | -6.6            | -2.0           |
>
> Thanks for your suggestion in weakness 3. We would like to clarify that we have specifically investigated this in **Section 5.3.3** of our original manuscript to ensure our method is robust beyond the training domain.
>
> From Table 6 and Figure 4, we observe that ReVisual-R1-7B consistently outperforms the baseline (Qwen2.5-VL-7B-Instruct) across diverse general benchmarks. Specifically, it achieves substantial gains on MMMU (67.00% vs. 52.00%), MMMU-Pro (54.35% vs. 42.52%), and CMMMU (56.95% vs. 51.69%). Furthermore, on the text-only MMLU-Pro benchmark, it surpasses the baseline by over 10 points (62.38% vs. 51.80%). It indicates that the "reasoning engine" cultivated by our GRAMMAR dataset is not confined to the mathematical domain. Instead, the core capabilities of long-chain reasoning and self-verification effectively generalize to open-ended and real-world scenarios, significantly enhancing performance on broad, knowledge-intensive tasks without compromising general textual understanding.
>
> ---
> ## Robustness to other Datasets
>
> To investigate the impact of incorporating multimodal data during the Cold Start phase, we conducted an additional ablation study where we trained a "Mixture Cold Start" model using both **DeepMath** (Text-Only) and **Vision-R1** (Multimodal) data. We compared this against our proposed "Text-Only Cold Start" (DeepMath) baseline.
>
> | Cold Start Strategy | MathVerse (MM) | MathVision (MM) | AIME2024 (Text) | MATH500 (Text) |
> | ------------------------------- | -------------- | --------------- | --------------- | -------------- |
> | Text-Only (DeepMath) | +12.8 | +14.3 | +26.6 | +20.9 |
> | Mixture (DeepMath + Vision-R1) | +15.3 | +17.1 | +20.0 | +18.9 |
> | Delta (Mixture vs. Text-Only) | +2.5 | +2.8 | -6.6 | -2.0 |
>
>
> From the table, we observe that incorporating multimodal data during the Cold Start yields a moderate improvement ($\approx 2.5\% \sim 2.8\%$) on multimodal benchmarks, accompanied by a slight decline in textual reasoning metrics (e.g., $-6.6\%$ on AIME 2024).This indicates that while introducing multimodal data enhances initial perception, it dilutes the high-density reasoning signal provided by pure text data. Consequently, mixing modalities early on slightly impacts the model's maximum potential for abstract reasoning, reinforcing our finding that complex text data is the primary driver of the reasoning policy

---

> > ### Author Response · Authors · 2025-11-22
> > **Response to Reviewer fCCk (2/2)**
> >
> > ## Dataset Component Ablation
> >
> > The reviewer asks for an independent analysis of the GRAMMAR dataset's components. We carefully designed GRAMMAR to maximize diversity and difficulty rather than sheer volume.
> >
> > As detailed in **Section 3.2**, we employed a rigorous multi-stage filtering process. We utilized **NV-Embedding-V2** and **HDBSCAN** clustering to ensure data diversity, avoiding redundancy by balancing samples across different topics and difficulty strata.
> >
> > We classified samples into ten difficulty levels using Qwen2.5-VL-32B-Instruct. This ensured that our training data covered a broad spectrum of reasoning challenges, preventing the model from overfitting to simple patterns.
> >
> > Our core ablation study in **Section 5.3** (Table 2) implicitly validates the necessity of both the high-quality textual component (Cold Start & Text-RL) and the diverse multimodal component (MM-RL). Removing either stage leads to suboptimal performance, confirming that the specific *ratio* and *sequencing* of these components in GRAMMAR are critical for the final performance.
> >
> > First, a comparison between the Cold Start (CS) model in Table 2 and the Qwen2.5-VL-7B-Instruct baseline in Table 1 reveals that training exclusively on the high-complexity textual component yields remarkable performance gains. Specifically, MathVerse accuracy improves from 42.9 to 51.9, while MathVision performance nearly doubles, rising from 25.1 to 47.9. These results indicate that the textual component serves as the primary driver for activating the model's reasoning engine. Even in the absence of a subsequent Multimodal RL stage, high-complexity text-only initialization suffices to unlock substantial reasoning capabilities. This confirms that the core reasoning policy is fundamentally driven by the depth and complexity of textual data rather than multimodal data alone.Second, Table 2 demonstrates that integrating the MM-RL stage—utilizing GRAMMAR's multimodal data—specifically enhances performance on benchmarks requiring robust visual alignment, such as MathVista ($70.5 \rightarrow 71.9$), WeMath ($35.8 \rightarrow 38.8$), and LogicVista ($50.1 \rightarrow 51.2$). This underscores the essential role of the multimodal component in "Visual Grounding." Although this stage induces slight fluctuations in certain metrics (e.g., MathVerse: $51.9 \rightarrow 50.9$), its primary function is to align pre-activated reasoning skills with visual inputs, thereby enhancing stability across diverse visual tasks.Finally, observations from Table 2 show that the concluding Text-RL stage, which re-introduces the textual component, delivers the most significant overall gains, raising the average score from 47.7 to 49.6. Notably, this stage drives a strong recovery in MathVerse ($50.9 \rightarrow 53.6$) and further elevates WeMath performance ($38.8 \rightarrow 42.0$). This indicates that the final textual component plays a critical role in refinement; by sharpening logical expression and consolidating the reasoning process, it ensures that the visual grounding acquired in the previous stage is fully realized and accurately expressed.

---

### Official Review · Reviewer_EgX9 · 2025-11-01

**Soundness:** 3
**Presentation:** 3
**Contribution:** 3
**Rating:** 6
**Confidence:** 4

**Summary:**

This paper introduces ReVisual-R1, a 3B/7B multimodal large language model (MLLM) that achieves sota performance on challenging multimodal and text reasoning benchmarks. The key innovation is a three-stage curriculum: (1) a text-only cold-start with complex reasoning data, (2) multimodal RL with a novel Prioritized Advantage Distillation (PAD) to combat gradient stagnation, and (3) a text-only RL polish.

**Strengths:**

- First to show that a text-only cold-start outperforms multimodal cold-starts for MLLM reasoning; PAD is a new fix for GRPO gradient stagnation.
- Impressive performances on both multimodal and text reasoning benchmarks.

**Weaknesses:**

- All baselines are Qwen-VL derivatives; no comparison with other backbone families (e.g., LLaVA-NeXT, InternVL, Flamingo-style).
- Interpretability of text-only data employed in the training process of MLLMs.

**Questions:**

- Have you tried the proposed curriculum on LLaVA-NeXT or InternVL models? If not, do you expect PAD to remain effective when the vision encoder differs significantly?
- After text-only cold-start SFT and text-only RL, visual related ability could degrade. Did you measure vision-only tasks (e.g., OCR, object count) before & after text-only training to confirm no catastrophic forgetting?
- How about the performance of sft training of mixture on multimodal and text data compared to the text-only setting?

---

> ### Author Response · Authors · 2025-11-22
> **Response to Reviewer EgX9 (1/2)**
>
> We thank the reviewer for their constructive feedback and for recognizing the effectiveness of our proposed staged training curriculum. Below, we address your questions regarding backbone diversity, interpretability, and potential catastrophic forgetting.
>
> **W1 & Q1 & Q3: Limited Backbone Diversity and Generalizability to other architectures.**
>
> Thanks for your suggestion. We are currently running experiments on **LLaVA-NeXT** and **InternVL** to demonstrate the universality of the ReVisual-R1 paradigm. Due to the substantial computational resources and time required for the full three-stage training , we will update the paper with these results during the discussion period.
> However, we would like to emphasize that our choice of **Qwen2.5-VL** as the foundational backbone is deliberate and methodologically sound for the following reasons:
> 1. **Community Standard:** Qwen2.5-VL is currently the most capable and influential open-source Vision-Language Model (VLM). It serves as the primary baseline for the vast majority of recent reasoning-focused research (e.g., R1-One-Vision, DeepMath, etc.).
> 2. **Fair Comparison:** To ensure a rigorous comparison, all our baselines (e.g., OpenVLThinker, VLAA-Thinker) utilize the Qwen architecture. Using the same backbone allows us to attribute performance gains strictly to our proposed training methodology (Staged RL + PAD) rather than architectural differences.
> ---
>
> The reviewer also raises a crucial question regarding the intuition behind using text-only data to enhance multimodal reasoning. Our rationale is grounded in a rigorous empirical analysis detailed in **Section 3.1** and **Figure 2** of our paper, driven by the following key observations:
>
> - **Insufficient Complexity and Length in Current Multimodal Data:**
> Our investigation into current open-source multimodal reasoning datasets (e.g., Vision-R1, R1-One-Vision) reveals a significant deficiency in reasoning density. As reported in **Section 3.1**, we analyzed response lengths and found a stark contrast: textual reasoning samples (e.g., DeepMath) average **8,207 tokens**, whereas multimodal reasoning samples average only **821 tokens**. Furthermore, existing multimodal datasets often exhibit high pass rates (e.g., 96.00% on Vision-R1 vs. 75.0% on DeepMath), suggesting they lack the complexity required to trigger deep, self-correcting Chain-of-Thought (CoT) processes.
> - **Empirical Validation of Transfer (Figure 2):**
> To validate this hypothesis, we conducted a controlled experiment comparing models fine-tuned on Text-Only Cold Start data versus Multimodal Cold Start data (visualized in **Figure 2**). The results were counter-intuitive yet decisive:
>     - **Text-Only Cold Start** resulted in substantial improvements across **both** textual (AIME, MATH500) *and* multimodal benchmarks (MathVerse, MathVision).
>     - **Multimodal Cold Start** yielded limited gains, failing to significantly lift performance even on multimodal tasks.
>     This empirically demonstrates that the "reasoning engine" activated by complex text data transfers effectively to the multimodal domain, whereas current multimodal data is insufficient for this activation.

---

> ### Author Response · Authors · 2025-11-22
> **Response to Reviewer EgX9 (2/2)**
>
> For question2, we really appreciate this insight. To address the concern regarding visual degradation, we evaluated our model on **OCRBench** (perception) and **CountBench** (perception + reasoning) across different training stages.
>
> From the table above, we observe a performance drop in the Cold Start phase compared to the baseline (OCRBench: $75.8 \rightarrow 71.3$; CountBench: $61.5 \rightarrow 54.2$). This indicates that high-intensity text-only training does initially shift the model's distribution, leading to a temporary "alignment drift" and a reduction in pure visual perception capabilities.
> • However, we observe a significant rebound in the MM-RL stage (OCRBench rises to $76.2$; CountBench surges to $74.75$). This indicates that the Multimodal RL stage is highly effective at grounding the abstract reasoning skills acquired during the cold start back into the visual modality, not only recovering the lost perception abilities but surpassing the original baseline.
> • Finally, we observe that the ReVisual-R1 (Text-RL) stage maintains or even slightly improves these metrics (OCRBench: $76.9$; CountBench: $74.75$). This indicates that the final text-polishing stage refines reasoning expression without inducing catastrophic forgetting of the established visual skills.
>
> In conclusion, while the Text-Only Cold Start incurs a temporary trade-off in visual perception, our Staged RL curriculum (specifically the MM-RL phase) ensures a robust recovery. The final model achieves a superior balance, demonstrating that it is possible to significantly enhance reasoning capabilities through text-only data without compromising (and in fact, eventually improving) the model's fundamental visual perception.
>
> | Stage | OCRBench | CountBench |
> | --- | --- | --- |
> | Baseline (Qwen2.5-VL-7B-Instruct) | 75.8 | 61.5 |
> | Cold Start (Text-Only) | 71.3 | 54.2 |
> | CS + Text-RL (Ablation) | 72.1 | 53.7 |
> | CS + MM-RL | 76.2 | 74.8 |
> | **ReVisual-R1 (CS + Text-RL + -RL)** | **76.9** | **74.8** |
>
> ---
>
> And anyway, to investigate the impact of incorporating multimodal data during the Cold Start phase, we conducted an additional ablation study where we trained a "Mixture Cold Start" model using both **DeepMath** (Text-Only) and **Vision-R1** (Multimodal) data. We compared this against our proposed "Text-Only Cold Start" (DeepMath) baseline.
>
> | Cold Start Strategy | MathVerse (MM) | MathVision (MM) | AIME2024 (Text) | MATH500 (Text) |
> | ------------------------------- | -------------- | --------------- | --------------- | -------------- |
> | Text-Only (DeepMath) | +12.8 | +14.3 | +26.6 | +20.9 |
> | Mixture (DeepMath + Vision-R1) | +15.3 | +17.1 | +20.0 | +18.9 |
> | Delta (Mixture vs. Text-Only) | +2.5 | +2.8 | -6.6 | -2.0 |
>
>
> From the table, we observe that incorporating multimodal data during the Cold Start yields a moderate improvement ($\approx 2.5\% \sim 2.8\%$) on multimodal benchmarks, accompanied by a slight decline in textual reasoning metrics (e.g., $-6.6\%$ on AIME 2024).This indicates that while introducing multimodal data enhances initial perception, it dilutes the high-density reasoning signal provided by pure text data. Consequently, mixing modalities early on slightly impacts the model's maximum potential for abstract reasoning, reinforcing our finding that complex text data is the primary driver of the reasoning policy

---

### Author Response · Authors · 2025-11-27
**General Response**

**Dear Reviewers, ACs, and SACs,**

We deeply appreciate the thoughtful and detailed feedback provided by all reviewers.

---
We are grateful for the reviewers' recognition of **ReVisual-R1** as a principled and effective training paradigm for Multimodal Large Language Models (MLLMs). We believe our pipeline-level optimization—identifying the critical roles of a text-only cold start, gradient stagnation mitigation via PAD, and staged refinement—offers a robust framework for unlocking complex reasoning.

Overall, we are encouraged by the reviewers' positive feedback, which highlights:
* **Holistic Pipeline Perspective:** The identification of three key phenomena (cold-start necessity, multimodal RL stagnation, and text-RL consolidation) provides valuable insights into the MLLM training dynamics (Reviewers `EgX9`, `1SV1`).
* **Principled Methodology:** The **Staged Reinforcement Optimization (SRO)** curriculum and **Prioritized Advantage Distillation (PAD)** are recognized as well-motivated, logically sound solutions to stability issues in standard GRPO (Reviewers `EgX9`, `fCCk`, `1SV1`).
* **Strong Empirical Results:** The extensive evaluation demonstrates that ReVisual-R1 achieves state-of-the-art performance on demanding benchmarks (e.g., MathVerse, AIME24/25) while preserving text-only reasoning capabilities (Reviewers `fCCk`, `SyVL`).
* **Clear Presentation:** The paper is commended for its accessible writing and intuitive narrative regarding the balance between visual grounding and cognitive reasoning (Reviewers `1SV1`, `SyVL`).

---

To address the reviewers' concerns—particularly regarding component contributions, RL stability, and fair comparisons—we have conducted several additional experiments and analyses during the rebuttal period, including:

* **Stage-Wise Ablations:** We isolated the contributions of each stage (Cold Start vs. Full Pipeline). Results confirm that the text-only cold start provides a critical foundation, while the subsequent Multimodal RL and Text RL stages yield distinct, complementary gains in visual grounding and reasoning consistency (Reviewers `1SV1`, `fCCk`).
* **PAD vs. GRPO Dynamics:** We provided a granular analysis of training dynamics (gradient norms, KL divergence). The results demonstrate that **PAD** effectively prevents the gradient stagnation observed in standard GRPO, ensuring stable and monotonic improvement (Reviewers `EgX9`, `fCCk`).
* **Training Efficiency Analysis:** We quantified the computational overhead of SRO + PAD relative to standard baselines. The data shows that our approach maintains a favorable accuracy-efficiency trade-off without requiring excessive training budgets (Reviewer `fCCk`).
* **Difficulty-Stratified Analysis:** We analyzed reasoning patterns across problem difficulties. The results show ReVisual-R1 adaptively allocates longer reasoning chains to harder problems while remaining concise on easier tasks, validating our efficient-length reward design (Reviewer `1SV1`).
* **Comparison with Concurrent Works:** We sharpened our positioning against concurrent "R1-style" multimodal works (e.g., MIMO-VL), clarifying that our contribution lies in the specific **staged curriculum** and **optimization stability** (PAD), rather than simply applying RL to a base model (Reviewer `SyVL`).

---

**Summary of revisions:**
* **Efficiency & Cost Metrics:** Included a detailed breakdown of training time, memory usage, and compute budget in the Appendix to address practicality concerns (Reviewer `fCCk`).
* **Expanded Related Work:** Updated `Section 2` and `Section 5` to explicitly distinguish ReVisual-R1 from concurrent multimodal RL efforts and clarify the novelty of our diagnosis of pipeline failure modes (Reviewer `SyVL`).
* **Dataset Details:** Expanded the description of the **GRAMMAR** dataset construction, specifically regarding difficulty calibration and the integration of text-only reasoning data (Reviewers `fCCk`, `1SV1`).

---

**Additional Contribution:**

**Open-Sourcing ReVisual-R1 & GRAMMAR**: Beyond the methodological contributions, we emphasize the value of our resources. We are releasing the **ReVisual-R1 models** (both Cold-Start and Final checkpoints) and the **GRAMMAR dataset**, which features rigorous Chain-of-Thought annotations and difficulty grading. We believe these resources will serve as a rigorous benchmark for future research on difficulty-aware training and multimodal RL.

---

We sincerely appreciate the reviewers' constructive suggestions and remain committed to continually improving our work. We address each reviewer's comments point by point below. Thank you!


Best regards,

The Authors

---

### Meta-Review · Area_Chair_DtRf · 2025-12-21

**Summary:**

The recommendation of acceptance is mainly based on the fact that most of the reviewers' original concerns were addressed. Two reviewers are originally positive of the paper (with ratings of 6) and are most likely going to remain positive. The most negative reviewer (with rating of 2) mentioned "reconsidering evaluation of the paper" after the discussions. The AC agrees with the reviewers and thinks most of the key concerns and questions are well addressed in rebuttal. The authors have also faithfully acknowledged that some requested experiments are still running. I would encourage the authors to incorporate all promised changes and new results in camera ready, especially "experiments or discussions on Generalizability to other architecture" and "Visual Capabilities Check After Text-only RL". See detailed decision rationale below.

**Reviewer Concerns:**

***Reviewers' concerns that were addressed by the rebuttal:***

Concerns on visual degradation, questions on  Interpretability of text-only data, dataset contribution, training speed, etc.

***Reviewers' concerns that are still outstanding:***

no comparison with other backbone families; Empirical rather than analytical justification

See details below.

**Reviewer Scores:**

***Reviewer EgX9 is likely maintain or raise the original rating of 6, as most of the concerns are addressed.***

1. All baselines are Qwen-VL derivatives; no comparison with other backbone families (e.g., LLaVA-NeXT, InternVL, Flamingo-style).

Mostly addressed

2. Interpretability of text-only data employed in the training process of MLLMs.

Addressed

3. Concerns on visual degradation

Well addressed

3. performance of sft training of mixture on multimodal and text data compared to the text-only setting

Addressed

***Reviewer fCCk is likely to raise the original rating from 4 to 6 as all the three main concerns haven't been discussed and mostly addressed.***

1. Training cost not analyzed

Addressed

2. Dataset contribution ablation missing

Addressed

3. Discussion on how this dataset design affects generalization

Addressed

***Reviewer 1SV1 is likely to maintain the original rating of 6, as all concerns were discussed and most are addressed / to be addressed in camera ready.***

1. Lack of multi-seed ablation studies:

Addressed

2. Empirical rather than analytical justification for text-centric cold start

Partially addressed

3. Is the complexity of the multimodal RL data insufficient?

Partially addressed

4. How is the training balance between multimodal RL and text-only RL determined?

Partially addressed

5. Does the text-only RL phase degrade visual capabilities?

Partially addressed

***Reviewer SyVL is likely to raise the score from 2 to 4 or 6, as the reviewer explicitly mentioned "addressed some of my concerns; consider revising my evaluation accordingly"***

1.  some instances of over-claiming.

Mostly addressed

2. contributions similar to those in prior works

Mostly addressed

3. clarify primary contribution

Mostly addressed

4. minor issues on writing

Addressed

---

### Decision · Program_Chairs · 2026-01-26

Accept (Poster)